# Spatially non-uniform condensates emerge from dynamically arrested phase separation

Nadia A. Erkamp[1,10], Tomas Sneideris[1,10], Hannes Ausserwöger [1], Daoyuan Qian [1], Seema Qamar[2], Jonathon Nixon-Abell [2], Peter St George-Hyslop [2,3,4], Jeremy D. Schmit [5], David A. Weitz [6,7,8] & Tuomas P. J. Knowles [1,9] ✉

The formation of biomolecular condensates through phase separation from proteins and nucleic acids is emerging as a spatial organisational principle used broadly by living cells. Many such biomolecular condensates are not, however, homogeneous fluids, but possess an internal structure consisting of distinct sub-compartments with different compositions. Notably, condensates can contain compartments that are depleted in the biopolymers that make up the condensate. Here, we show that such double-emulsion condensates emerge via dynamically arrested phase transitions. The combination of a change in composition coupled with a slow response to this change can lead to the nucleation of biopolymer-poor droplets within the polymer-rich condensate phase. Our findings demonstrate that condensates with a complex internal architecture can arise from kinetic, rather than purely thermodynamic driving forces, and provide more generally an avenue to understand and control the internal structure of condensates in vitro and in vivo.

It is crucial for life to have spatial and temporal control over the biochemical reactions it performs[1,2]. Membraneless organelles in the form of biomolecular condensates are an important tool to organise proteins and nucleic acids and they perform a range of different functions, including storage[3], processing[4], signalling[5] and modulating gene expression[6]. Crucially, many functional condensates, such as stress granules[7], nuclear speckles[8] and the nucleolus[9], are multiphase, they consist of sub-compartments containing distinct compositions of proteins and nucleic acids[10–13]. Spatially structured condensates were shown to arise in minimal mixtures[13]. By creating additional compartments within the condensate, cells have further control over the interactions and reactions taking place within. Notably, sub-compartments that are poor in the biopolymers that compose the condensate have also widely been observed[14–18]. For example, "vacuole" formation in peptide-RNA condensates was described[15]. A similar "core-shell" structure has also been observed in microgels[19]. Moreover, in living cells, TDP-43 rich droplets with nucleoplasm-filled vacuoles[20], nuclear and cytoplasmic germ granules with hollow centres[21–23], and poly-rA RNA nuclear condensates with low-density space have been observed[24]. While this class of structure is thus commonly encountered, it is not yet clear what the driving forces are for the formation of biopolymer-poor phases inside of the biopolymer-rich condensate. Here, we show that this structure forms due to a dynamically arrested phase transition when the condensate is out-of-

[1]Yusuf Hamied Department of Chemistry, Centre for Misfolding Diseases, University of Cambridge, Lensfield Road, Cambridge CB2 1EW, UK. [2]Cambridge Institute for Medical Research, Department of Clinical Neurosciences, University of Cambridge, Cambridge CB2 0XY, UK. [3]Department of Medicine (Division of Neurology), University of Toronto and University Health Network, Toronto, Ontario M5S 3H2, Canada. [4]Department of Neurology, Columbia University, 630 West 168th St, New York, NY 10032, USA. [5]Department of Physics, Kansas State University, Manhattan, KS 66506, USA. [6]Department of Physics, Harvard University, 17 Oxford Street, Cambridge, MA 02138, USA. [7]John A. Paulson School of Engineering and Applied Sciences, Harvard University, Cambridge, MA 02138, USA. [8]Wyss Institute for Biologically Inspired Engineering, Harvard University, Cambridge, MA 02138, USA. [9]Cavendish Laboratory, Department of Physics, University of Cambridge, J J Thomson Ave, Cambridge CB3 0HE, UK. [10]These authors contributed equally: Nadia A. Erkamp, Tomas Sneideris. ✉e-mail: tpjk2@cam.ac.uk

equilibrium, rather than due to thermodynamic factors[25]. Limited diffusion causes condensates to deviate from the binodal during composition changes, causing nucleation of dilute phase droplets inside of the condensates. The deviation from equilibrium can be induced by changes in the environment.

## Results and discussion

### A double-emulsion structure can form reversibly

We explore the mechanism behind the formation of this internal structure with a model system containing single stranded Poly-rA RNA (2100–10600 subunits or 700–3500 kDa) and PEG (average ~450 subunits or 20 kDa), which co-condensate[26]. These condensates don't significantly solidify over time and change composition with temperature, containing more Poly-rA at 20 °C than at 55 °C (Supplementary Fig. 1). We vary the temperature to determine the effect of changing the condensate composition on the formation of trapped droplets, liquids poor in Poly-rA and PEG inside the condensate (Supplementary Fig. 2). We observe these droplets in condensates at 20 °C, but not at 55 °C (Fig. 1a). This is the case for multiple cycles of heating and cooling to these temperatures, with a condensate of this size showing $100 \pm 5.6$ trapped liquids in the pictures (Supplementary Figs. 3 and 4). We observe the condensate during the temperature change to see how these liquids are formed and removed, starting with a condensate with droplets at 20 °C (Fig. 1B.1). When heating to 30 and 55 °C, we observe that droplets become smaller and completely disappear respectively (Fig. 1B.2–5 and Supplementary Fig. 5). After the droplets are completely removed, we lower the temperature of these condensates and trapped droplets form again (Fig. 1B.6). The trapped droplets grow with decreasing temperature (Fig. 1B.7–9). At 20 °C, we obtain condensates that look very similar to the ones before the temperature change (Fig. 1B.10). We conclude that trapped droplet formation is reversible and takes place during a composition change. Droplet formation could be a kinetic process or a thermodynamic process. If the formation of enclosed droplets is a kinetic process we would expect droplets to form at fast composition changes, whereas droplets would be also form at slow composition changes if it is a thermodynamic process.

### The formation of enclosed droplets is a kinetic process

We varied the rate at which condensates are cooled from 55 to 20 °C to probe whether trapped droplets originate from a thermodynamic or kinetic process. A condensate with a radius of 24 μm will contain only 1 trapped droplet after cooling slowly at 1 °C/min, but 24 droplets after cooling faster at 20 °C/min (Fig. 2 and number of trapped droplets reported in Supplementary Fig. 6a). Thus, depending on the rate of

composition change, we obtain a kinetic product, a condensate with many trapped droplets inside, or a more thermodynamically stable product, a condensate with less enclosed droplets. This observation implies that these droplets are formed when the condensate is unable to reach the thermodynamic equilibrium, in which case we would expect that the number of trapped droplets also depends on the size of the condensates. Smaller condensates might equilibrate faster with the surrounding dilute phase than large condensates. Indeed, a condensate with a radius of 14 μm does not form trapped droplets at any of the tested cooling rates and the condensate with a radius of 38 μm contains more droplets than the condensate with a 24 μm radius, confirming that the formation of enclosed droplets is a kinetic process (Supplementary Fig. 6b). Condensates with droplets inside might be higher in energy due to the additional surface area with the droplets, which increases the surface energy. If this kinetic product was indeed higher in energy, we could expect that over time the number of droplets in condensates will be reduced.

### Trapped droplets similar in composition to bulk dilute phase

We observe a condensate with trapped droplets over time (Fig. 3a, Supplementary Movie 1). Enclosed droplets move through the condensate and fuse with each other once coming into contact. Similarly, trapped droplets fuse with the dilute phase surrounding the condensate, indicating that the liquid in the cavities and surrounding dilute phase are similar. Over time, the number of trapped droplets decreases via these two mechanisms and the average radius increases with time (Fig. 3b, c). Fitting the data in Fig. 3b, c with classical reaction kinetics and Lifshitz-Slyozov-Wagner (LSW) theory[28], we see that this behaviour follows the expected scaling laws (Supplementary note 1). We study these fusion events and the fusion of the condensates further to confirm that the droplets are similar in composition to the surrounding dilute phase. Specifically, we study condensates fusing (Fig. 3d), trapped droplets fusing (Fig. 3e) and a trapped droplet fusing with the surrounding dilute phase (Fig. 3f). The aspect ratio, the length and width ratio of the two fusing droplets, over time can be fitted with the exponential decay function to find the characteristic fusion time $\tau$[6,29]. Figure 3b–d show this characteristic fusion time against the average diameter of the fusing liquids. The slope of this fitted line is the inverse capillary velocity, which is a function of the surface tension and the viscosity[6,27,29]. The viscosity is that of the dense condensate phase, which is the same for the 3 different fusion events we study. Thus, any differences in the capillary velocity we determine shows a difference in the surface tensions between the different interfaces. However, we find very similar values for the inverse capillary velocities at the same temperature, for the different fusion events. Thus, the surface tension

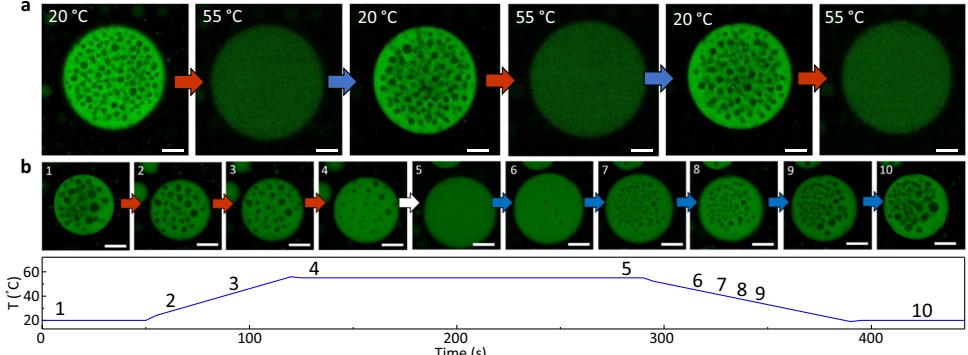

**Fig. 1 | Temperature-dependent and reversible trapped droplet formation.** **a** Confocal microscopy images of multiple cycles of trapped droplet formation and removal (Supplementary Fig. 3). Intensity differences are to show that the concentration of Poly-rA in the dense phase is higher at 20 than at 55 °C

(Supplementary Fig. 4). **b** Confocal microscopy images of temperature-induced morphology changes of Poly-rA PEG condensates. The intensity of the Poly-rA-rich phase is kept constant at every temperature and is thus not an indication of Poly-rA concentration. Poly-rA is fluorescently labelled. All scale bars represent 25 μm.

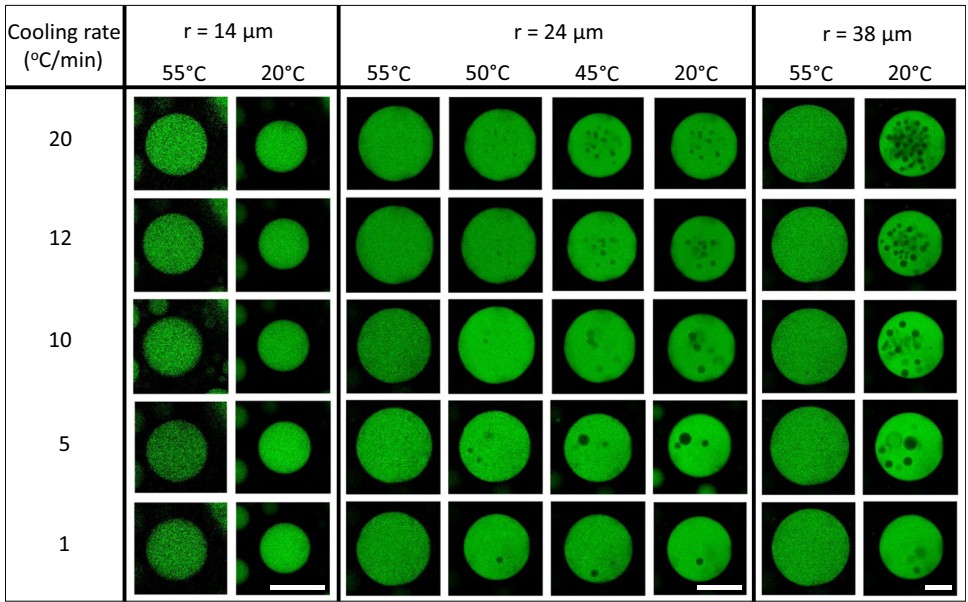

**Fig. 2 | Formation of trapped droplets depends on cooling rate and condensate size.** Confocal microscopy images of condensates, with a radius of 14, 24 and 38 μm at 55 °C, being cooled from 55 to 20 °C at different rates. The number of liquids trapped inside the condensate during cooling depends both on the size of the condensate and on the rate at which the condensates is cooled (Supplementary Fig. 6). Poly-rA is labelled. All scale bars represent 25 μm.

between the condensate and the outer dilute phase, and the condensate and the trapped droplet is very similar. This confirms that trapped droplets are very similar in content to the surrounding dilute phase. The trapped droplets are thus droplets of dilute phase trapped in the dense phase, rather than a third distinctly different phase.

### Double-emulsion condensates as a response to diverse changes

In the liquid-like Poly-rA PEG condensates, we have reversibly formed and removed dilute phase liquids by changing the temperature. However, we find that this structure can be obtained in diverse systems as a response to different changes. We prepared reconstituted stress granules, which are similar in composition to the condensates found in cells[30] (Fig. 4a and Supplementary Fig. 7). When additional nucleic acids are added to the solution, the stress granules rapidly take this up, changing their composition, and they then form trapped droplets. These trapped droplets contain, like the surrounding dilute phase, a low amount of nucleic acids and G3BP1. Thus, we observe that composition changes in this system also lead to the formation of a double-emulsion structure. Changes in compound concentrations in cells via active processes may thus induce the formation of double-emulsion condensates. Notably, the number of trapped droplets is higher in larger condensates (Fig. 4b), which matches with our previous observation that larger Poly-rA PEG condensates form more trapped droplets (Fig. 2 and Supplementary Fig 6a, b). In gels, cavities can be formed during synaeresis, in which a gel changes composition[31–33], for example due to a change in temperature[19], also supporting the idea that dynamically arrested phase transitions are a general way for producing double-emulsion structures. In addition, we have modelled how a phase separating system respond to changes in their environment (Fig. 5a, Supplementary Note 2). After formation, the condensates experience either a slow (top) or quick (bottom) change in composition and as a result form condensates without and with double-emulsion structure respectively (Supporting Movies 2, 3 and 4). Notably, larger condensates form more trapped droplets than smaller ones. Upon reversing the composition change, the trapped droplets are removed (Supplementary Movie 5). This model provides further highlights that the ability to form a double-emulsion structure is a general property of condensates undergoing quick composition changes.

### Dynamically arrested phase transition

Combining our observations allows us to understand the mechanism behind the formation of this double-emulsion condensates. Consider the binodal of a phase diagram for the Poly-rA PEG system, which shows the equilibrium concentrations in the dilute and dense phase (Fig. 5b). When a change in the environmental conditions occurs, the composition of the dilute and dense phase will change to their new equilibrium positions on the binodal. We have observed in multiple system that if a composition change occurs relatively quickly, droplets of dilute phase are formed in the condensate. We conclude that during the composition change, as the system moves from its original to its new equilibrium position (black dots), the system deviates from the binodal during this change (black lines) and droplets of the dilute phase can nucleate inside of the condensate (Fig. 5b, Supplementary Fig. 8 and Supplementary Note 3 and 4). The further the deviation the binodal during the trajectory, the more droplets of dilute phase can nucleate, matching observations that quicker changes in larger condensates result in the formation of more trapped dilute phase droplets. Interestingly, biomolecular condensates typically have a low surface tensions[34], which leads to a low energy penalty for droplet formation and thus a low nucleation barrier for forming a double-emulsion structure. Using the Poly-rA PEG system, we looked into why the system deviates from the binodal. To determine whether viscoelasticity could be a limiting factor, we measured the volume of a condensate when cooling at 1 and 20 °C/min. We found no significant difference between the volumes, indicating that at both cooling rates the equilibrium volume can be reached and viscoelasticity is unlikely the cause of the slow response (Supplementary Fig. 9). This matches with the liquid-like, rather than gel-like, nature of these condensates. To determine if diffusion of biopolymers in the condensate could be a limiting factor for equilibration during quick composition changes, we determined the diffusion coefficient of PEG in the condensates using fluorescence recovery after photobleaching (FRAP)[35] (Supplementary Fig. 10a, b). In addition, we determined if condensates of different sizes at different cooling rates form a double-emulsion structure (Fig. 4c). From this figure, we can see that there is a well-defined critical size for the formation of these droplets. We fit the critical sizes using the scaling law $D_{eff} = \frac{R^2}{t}$ derived from the diffusion equation[36,37], in which R

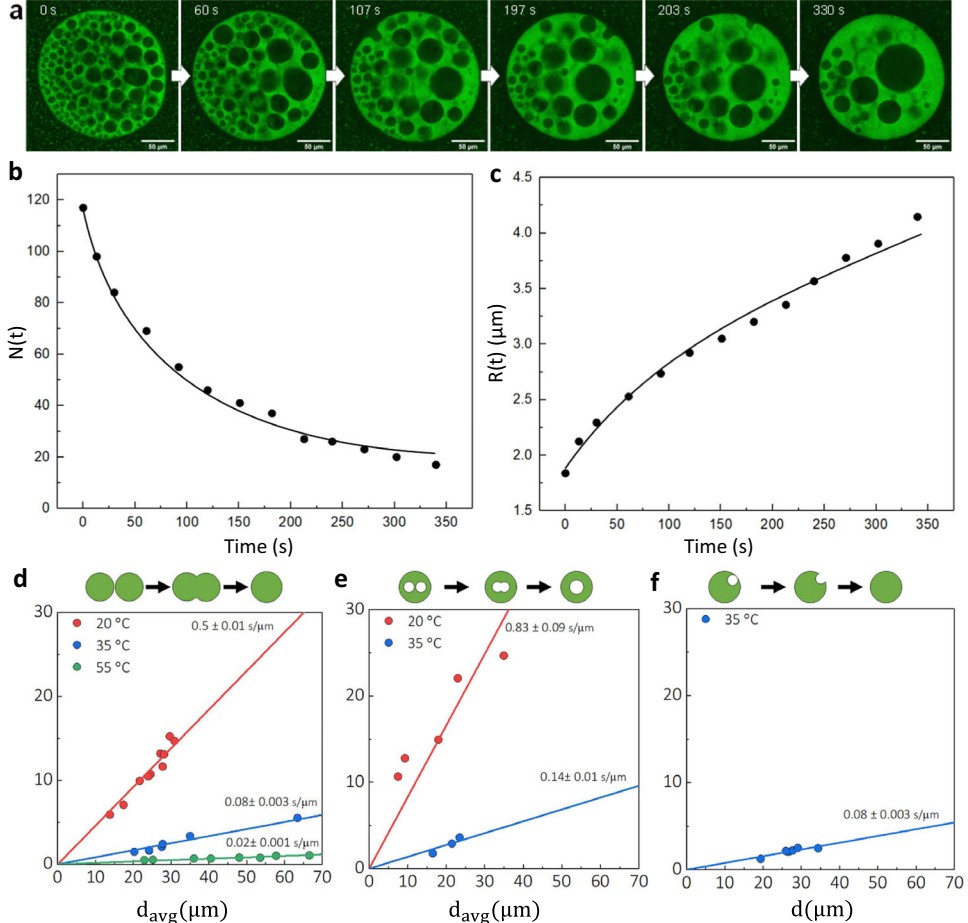

**Fig. 3 | Dynamics of trapped droplets and merging events. a** Trapped droplets inside of Poly-rA PEG condensates can merge with each other and with the outer dilute phase. The sample was incubated at 35 °C. Poly-rA is shown, the scale bar shows 10 µm. **b** The number and **c** average radius of the trapped droplets is shown over time, which follows the expected scaling laws (Supplementary Note 1). From

the fusing of **d** condensates, **e** trapped droplets and **f** droplets with the outer dilute phase, we extract the characteristic merging time τ. The slope of the plot τ against the average diameter of merging droplets is the inverse capillary velocity, which is approximately equal to the viscosity over the surface tension[6,27].

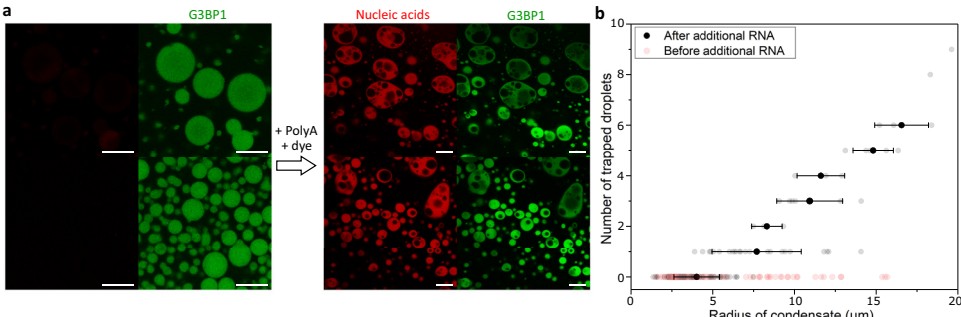

**Fig. 4 | Formation of double-emulsion reconstituted stress granules.
a** Reconstituted stress granules can form a double-emulsion structure when their composition changes upon the addition of 250 ng/µL Poly-rA RNA. The trapped droplets are poor in the protein G3BP1 and in nucleic acids. The before and after images show different condensates. **b** The number of trapped droplets in the

images of Fig. 4a and Supplementary Fig. 7 as function of the radius of the condensates. Similar to the Poly-rA PEG system, more trapped droplets can be formed in larger condensates. n = 157 and 104 for before and after additional or RNA respectively. The average number of trapped droplets and one standard deviation is shown after additional RNA.

is the diffusion length scale or radius of the condensate and $t = \frac{\Delta T}{v}$, where $v$ is the cooling rate and $\Delta T \approx 5$ °C is the amount of temperature change needed to form dilute phase droplets. Fitting this scaling law gives us an effective diffusion coefficient of approximately 12 µm²/s, which is a very similar value to the diffusion coefficients measured using FRAP. We conclude that the nucleation of dilute phase droplets

in the condensate can be explained completely by the slow diffusion of macromolecules in liquid-like biomolecular condensates. Condensates in cells often have a higher viscosity and thus lower diffusion coefficient[34], which means that the critical size or nucleation barrier to form double-emulsion condensates is lower (Supplementary Fig. 10c). Notably, condensates of small sizes require significantly larger or

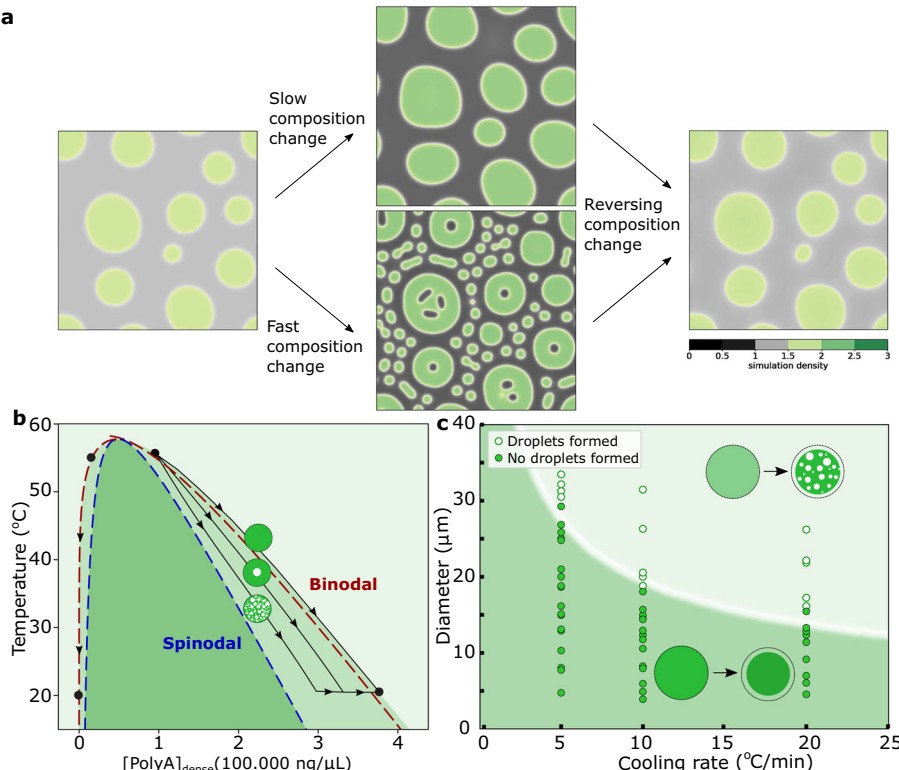

**Fig. 5 | The formation of double-emulsion condensates. a** Simulations of rapid or slow composition changes provide evidence that the ability to form double-emulsion structures upon fast composition changes is a general. (Supplementary note 2, Supplementary movies 2, 3, 4 and 5). **b** Poly-rA PEG condensates change composition when they are cooled from 55 to 20 °C. Depending on factors like the rate of the change and the size of the condensate, the condensate can follow the binodal quite closely or deviate from the binodal, at which point biopolymer-poor droplets can nucleate inside of the condensates (Supplementary Fig. 1a and 8, Supplementary note 3, 4). **c** Experimental data on whether or not condensates of a certain size nucleate droplets of dilute phase droplets (points) are fit with a scaling law to give an effective diffusion coefficient of $12\,\mu m^2/s$ shown as the curve in the background. The ability to from droplets depends strongly on the size of the condensate and the rate of composition change.

faster composition changes to from a double-emulsion structure than larger condensates (Supplementary Fig. 10d and Supplementary Note 5). Additionally, surface effects may play an important role in the ability of smaller condensates to form this structure. We expect double-emulsion condensates in cells to form most often in larger, viscous condensates undergoing significant composition changes. In general, dilute phase droplets nucleate inside of the condensate, because diffusion is limiting the rate at which composition changes can be achieved by the condensate responding to environmental changes.

Biomolecular condensates, including those found in cells, often contain distinct sub-compartments with different compositions. Sub-compartments poor in the biopolymers that make up the condensate are common, but the mechanism underlying their formation was not yet known. We have shown that double-emulsion condensates are formed due to a composition transition with a slow response leading to a dynamically arrested phase transition. In condensates, slow diffusion of biopolymers causes the condensate to deviate from the binodal, causing the nucleation of biopolymer-poor droplets in the condensate. This mechanism allows us to understand and control the formation biopolymer-poor liquids inside of condensates using a kinetic framework.

# Methods
## Materials
Poly-rA (MW 700–3500 kDa), PEG (average MW ~20,000), HEPES, glacial acetic acid, NaCl, KCl, ethanol and mPEG(5k)silane were obtained from Sigma Aldrich. Alexa Fluor™ 647 Carboxylic Acid and

RNAase free water was obtained from Thermo Fisher. BactoView™ nucleic acid binding dye (500×) was purchased from Biotium. PEG(20k)-AF647 was purchased from Nanocs. Sylgard 184 Elastomer base and curing agent were bought from Dow Corning Corporation. A total of $18 \times 18$ mm glass slides were purchased from Academy. $24 \times 60$ mm No.1.5 glass slides were obtained from DWK Life Sciences. $(U)_{40}$-Cy3 and $(U)_{40}$-Cy5 for FRET-FLIM were obtained from GenScript.

## Fabrication of imaging wells for confocal microscopy
Holes of 5 mm in diameter were punched in PDMS slabs of ~3 mm height using biopsy puncher. These slabs were plasma bonded to $24 \times 60$ mm No.1.5 cover glass slides to create wells. Both the wells and $18 \times 18$ mm glass slides used to seal the top of wells were treated with PEG-silane, using a method based on previously reports[13,14]. Briefly, treatment solution was made by mixing 10 mg of PEG(5000)silane with 20 uL of glacial acetic acid and 1 mL of ethanol. Wells and glass slides were treated by placing them in solution for 1 h at 65 °C and afterwards washing them thoroughly with water. Treated devices are not used after more than 3 weeks since the treatment.

## Making condensates
Poly-rA PEG condensates are prepared by mixing stock solution such that we obtain 2000 ng/uL PolyA, 2.5× BactoView™ nucleic acid binding dye, 5 w/w% 20.000 PEG (optionally, of which 0.1% is 20.000 PEG-AF647), 750 mM KCl, 50 mM HEPES at pH = 7.3 in RNAase free water. The Poly-rA concentration is determined with a nanodrop machine by measuring the absorbance at 260 nm before the addition of the dye.

Reconstituted stress granules were prepared as described previously[15]. Briefly, stress granules were formed by mixing cell lysate containing G3BP1-EGFP with recombinant G3BP1, resulting in a solution with 30 µM recombinant G3BP1 and 1 mg/mL protein concentration from the cell lysate. In our experiments, 250 ng/µL Poly-rA and 2.5 x BactoView™ dye is added. Notably, BactoView™ stains all nucleic acids, not just the Poly-rA.

## Confocal imaging

A Leica Stellaris 5 confocal microscope (confocal fluorescence imaging) (white light laser) microscope equipped with a 10× Nikon 0.3 NA or a 63× oil immersion Leica 1.4 NA is used for imaging. A TS102SI Instec rapid heating and cooling stage is used to control the temperature and change it at desired rates. Qualitative confocal images like the ones in Fig. 1 are taken in analogue mode. The intensity is then compared to a reference sample, containing Poly-rA, dye and PEG, to control for the influence of temperature on the brightness of the sample. A similar reference sample is used for the quantitative studies (Fig. 2), however then the photon counting mode was used. FRET-FLIM (Supplementary Fig. 1b) was performed using the TauSense mode. At each condition, the determined lifetime was average of the lifetime in condensates of at least 10 images. The standard deviation of the lifetimes for the pixels within an image was of less than 0.03 ns. Both the lifetime of just the donor and the donor in presence of the acceptor are measured at each temperature to determine the FRET efficiency E. To determine the temperature of the sample in comparison to the temperature of the temperature control stage (Supplementary Fig. 5), we measured the intensity of a solution of AF647 (10 uM) in the presence of Poly-rA PEG condensates. The intensity of the AF647 scaled linearly with temperature and was thus used to determine the temperature of samples during cooling. For FRAP (Supplementary Fig. 9), a 488 agon laser at 100% power is used to bleach disk of different area sizes. The recovery of fluorescence over time in the FRAP kymograph is fitted with a single exponential to obtain the half life time τ using the built-in software.

Only the images in Fig. 4 and Supplementary Fig. 7 were taken on an LSM 880 (Zeiss) microscope by exciting using an Argon multi-line 35 mW 488 nm laser (3 mW max at focal plane). The resulting fluorescence was collected using a 63× Plan-Apochromat 1.4 NA oil objective (Zeiss) and detected on a 34-channel spectral array detector in the 480–580 nm range. Samples were imaged with a Definite Focus module (Zeiss) employed for thermal drift correction and ZEN Black v2.3 (Zeiss) software used for acquisition.

Further analysis of pictures was performed using Fiji. The fitted lines of Fig. 3d–f are set to go through 0,0 since droplets of 0 µm fuse in 0 seconds.

## Production of G3BP1

Plasmid containing His-Sumo tagged G3BP1-FL was transformed into competent *E. coli* BL21 (DE3) (NEB, authenticated via STR profiling). A single colony was used to inoculate 5.0 mL of LB media containing kanamycin and incubated overnight at 37 °C. The starter culture was subsequently used to inoculate 10 litres of LB media containing kanamycin and incubated at 37 °C until the Absorbance at 600 nm reached 0.6 at which point the G3BP1 expression was induced by 0.6 mM IPTG and the cultures were incubated at 16 °C overnight. Cells were harvested by centrifugation at $3075 \times g$ for 20 min. Cell pellet was resuspended in Buffer A (50 mM Tris, 200 mM NaCl, 1 mM DTT pH = 8.0) plus protease inhibitor cocktail and subjected to high pressure cell lysis using Constant Pressure Cell Disruption System (Constant systems. Ultracentrifugation at $100,000 \times g$ was employed to clarify the cell lysate prior to loading onto a 20 mL Ni-Sepharose Advance resin containing gravity column (Bioserve). His-SUMO tagged G3BP1 was purified using standard Ni-affinity protein purification protocol which included a wash step in Buffer A containing 25 mM Imidazole and an elution step in Buffer A containing 500 mM Imidazole. The column eluates were run on an SDS-PAGE and the fractions containing the protein were pooled, mixed with His-tagged ULP protease for the cleavage of the His-SUMO tag and dialysed in buffer A overnight at 4 °C. Cleaved protein was further treated with 0.1 mg/mL RNase and DNase by incubating it at 37 °C for 15 min. Post incubation, protein sample was diluted using Buffer A without NaCl to lower the salt concentration to 50 mM before loading it on a SP-Sepharose (Cytiva) ion-exchange column. Protein was further purified by running a salt gradient and the fractions containing the protein were pooled, concentrated and subjected to the gel filtration step using a Superdex-200 Increase column (Cytiva) in the storage buffer (50 mM HEPES, 400 mM NaCl, 1 mM DTT, pH = 7.5). Protein fractions containing G3BP1 were assessed by SDS-PAGE and the pure fractions were pooled and concentrated to 15 mg/mL. Protein was aliquoted and snap frozen in Liquid Nitrogen for all subsequent assays.

## Cell Line Generation for G3BP1-mEmerald cell lysate

To generate the G3BP1-mEmerald HeLa line, a PiggyBac transposon-transposase system was used. Initially, a vector was designed using Gibson assembly of the G3BP1 transgene into a custom mEmerald-PiggyBac vector (available upon request). Cotransfection of the G3BP1-mEmerald-PiggyBac vector and a transposase (TransposagenBio) plasmid into CCL-2 HeLa cells (ATCC, authenticated via STR profiling) was followed by blasticidin selection to isolate a polyclonal G3BP1-mEmerald population. Transfections were performed using FuGene (Roche) at a 1 µg DNA: 3 µl FuGene ratio. Cells were maintained at 37 °C and 5% $CO_2$ in Dulbecco's Modified Eagle Medium (DMEM, Gibco) supplemented with 10% foetal bovine serum (FBS, Gibco).

## Generation of G3BP1-mEmerald cell lysate

The generation of cell lysates from G3BP1-mEmerald HeLa cells and the subsequent generation of lysate granules have been described previously[15]. Briefly, cells were grown to 100% confluency in a 10 cm dish and harvested in 5 mL PBS. All steps were performed at room temperature. Collected cells were spun at $500 \times g$ for 5 min with the supernatant aspirated and discarded. Cell pellets were stored at −80 °C until ready to use. To prepare cell lysates, pellets were thawed for 2 min and resuspended in 250 µL lysis buffer containing 50 mM TRIS pH7.0 (Sigma), 0.5% NP40 (Thermo), 0.025× mini protease inhibitor tablet (Roche) and 40× murine RNase inhibitor (NEB). After a 3-min incubation period lysates were spun at $24,000 \times g$ for 5 min to remove nuclei and cell debris. The supernatant was retained to produce reconstituted stress granules when mixed with unlabelled G3BP1.

## Statistics and reproducibility

Figure 1a, b show representative images of the condensates undergoing a temperature change. The temperature change was repeated 10 times (images of 1 condensate for 6 cycles shown in Fig. 1a and supplementary Fig. 3). Similar observations were made in the other condensates (>100) in the sample. Fig. 2, Fig. 3a supplementary Movie 1 and supplementary Fig. 2, supplementary Fig. 6b, c show representative images or movies. Similar observations in other condensates (>100) with a different size in the sample were made. The experiment in Fig. 4a was repeated three times and similar results were obtained (pictures included in supplementary Fig. 7). Supplementary Fig. 10b shows a representative example of the experiments which each contribute a datapoint to Supplementary Fig. 10a. 24 FRAP experiments were performed.

## Reporting summary

Further information on research design is available in the Nature Portfolio Reporting Summary linked to this article.

## Data availability

Further data generated in this study are provided in the Supplementary Information and source data file available online. Other data generated during the study are available on reasonable request from the corresponding author: tpjk2@cam.ac.uk.  Source data are provided with this paper.

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

## Acknowledgements

We thank Prof. Frans Spaepen and Prof. Marina Feric for discussions. The research leading to these results has received funding from the Royall Scholarship (N.A.E.), the European Union's Horizon 2020 research and innovation programme under the Marie Skłodowska-Curie grant MicroREvolution (agreement no. 101023060; T.S.), Global Research Technologies Novo Nordisk A/S (H.A., T.P.J.K.), a Henry Wellcome Fellowship (218651/Z/19/Z, J.N.A.), Canadian Institutes of Health Research (Foundation Grant and Canadian Consortium on Neurodegeneration in Aging Grant, P. St G.H.), Wellcome Trust Collaborative Award 203249/Z/16/Z (P. St G.H., T.P.J.K.), and US Alzheimer Society Zenith Grant ZEN-18-529769 (P. St G.H.), the NIGMS (R01GM141235; J.D.S.), the European Research Council under the European Union's Seventh Framework Programme (FP7/2007-2013) through the ERC grants

PhysProt (agreement no. 337969; T.P.J.K.) and the Newman Foundation (T.P.J.K., T.S.).

## Author contributions

N.A.E., T.S., J.S., H.A. and T.P.J.K. conceived the study. N.A.E., T.S., H.A., D.Q., J.N.-A., J.D.S. and D.A.W. performed investigation. S.Q., T.P.J.K. and P.St.G-H. provided resources. T.P.J.K. and P.St.G-H. acquired funding. N.A.E., T.S. and T.P.J.K. wrote the original draft, all authors reviewed and edited the paper.

## Competing interests

The authors declare no competing interests.
