## [Peer Review File · Nature Communications]

Editorial Note: Parts of this Peer Review File have been redacted as indicated to remove third party material where no permission to publish were obtained

REVIEWER COMMENTS

Reviewer #1 (Remarks to the Author):

In this study by Erkamp, Sneideris et al., the authors propose that kinetic arrest could contribute to formation of condensates with cavities, which they term “multiphase” condensates. They develop a model system of RNA and PEG phase separation and tunably induce cavity/vacuole formation through temperature sweeps. Through a clean assay modulating the importance of kinetics vs thermodynamics by changing the rate of temperature change, they identify features that contribute to cavity stability and size. They identify that these cavities resemble the dilute phase and undergo merging and growth. Overall, they propose that the quick quenches move the system closer to the spinodal, which due to the slower diffusion of biomolecules in the dense phase, leads to nucleation of vacuoles within the dense phase. While they develop their assays with temperature quenches, they hypothesize the generality with different quench/annealing parameters that control the importance of kinetics i.e. including pH, salt, and composition.

Overall, the paper is compelling and an interesting study. The development of their assay is well-motivated and by large, the conclusions are well-supported by the experiments. I outline a few concerns/changes - intended to clarify presentation, improve quantification and/or reanalyses of existing data-sets to strengthen existing claims, and re-evaluate certain experiments. If addressed, I believe the manuscript will be improved and would recommend publication.

Concerns:

1. Terminology: The authors use the term “multiphase” condensate broadly to discuss their specific example - of dilute phase nucleation arising from kinetic arrest in the dense phase. I am puzzled by this choice of wording as one is led to believe there are more than 2 phases existing in systems - indeed, this is not the case. Rather, rapid shifts in composition lead to heterogeneous nucleation within dense phases, a phenomenon known before in different contexts. As the authors motivate themselves, the literature references include 2 distinct phenomenology → examples like the nucleoli or paraspeckles are multiphase in the sense that their structure is characterized by more than one order-parameter (depending on specific combination of compositions in each phase) and a secondary case with vacuoles or cavities (especially for the system motivated here - a single equilibrium order parameter suffices). I believe that this essential issue needs to be adequately addressed:

a. Considering rewording description of their system to “vacuole” or “cavity” or “heterogeneous” or an equivalent example the authors may come-up with to replace multiphase.

b. In their introduction, the authors directly jump from “multiphase subcompartments” (which are not biopolymer poor, often different in relative concentrations and distinct from nucleoplasm) to “vacuoles” and “core-shell” architectures. I would recommend clarifying that the “specific” examples listed constitute cases where the internal subcompartments of vacuoles or cavities are biopolymer depleted.

2. Quantification of cavity sizes and number: While the visual image for cavities changing as a function of temperature are appealing, statements that comment on cavities growing smaller or larger and reversibly would be better served by quantification. To this end, I recommend incorporating analyses/text that directly addresses:

a. Incorporating a condensed data-plot similar to Ext Data Fig 6 in Figure 2 that shows the variation in cavity number, cavity size etc. with different condensate sizes and cooling rates.

b. One expects the nucleation rate to scale non-linearly with super-saturation. Is the cooling rate a proxy for super-saturation? If so, is Ext. Data 6 consistent with this? Can the authors also specify what time-point the data is reported for number?

c. Is there a reproducibility in the number or size of cavities? For e.g. do the numbers average for the same-condensate across multiple, well-separated temperature cycles? How about for 2 condensates of similar sizes?

Including such analyses more broadly, including for Ext Data Fig 3. should be fairly straightforward and will improve interpretation.

3. Cavity dynamics: Motivated by (2), it would be interesting to report on dynamics of condensate sizes. For e.g., in the data reported in Fig. 3 → does the cavity formation have any resemblance to classical nucleation/growth w.r.t scaling laws or distributions in size?

I note here that no new analyses is strictly required and reporting the variations as a function of time would be sufficient, although the authors may choose to add-in more along the lines suggested outlined above.

4. Figure 4A, while motivated in good spirit, is a little confusing with binodals/spinodals since T is used as the experimental knob throughout this study. I recommend either changing or including a version specifically with temperature (like the Ext. Data Fig)

5. As written the "slow diffusion" section is a little dense. Can the authors, by text or otherwise, clarify the logic behind the diffusion limited argument i.e to show that concentration increases require diffusion across "length-scales" measured by $\langle x \rangle$ and thus droplets larger than that compensate for this change through nucleation of dilute phases? If the reviewer misunderstood the argument, clarity by rewriting text could only help.

6. The last section on kinetic arrest in diverse systems is confusing and unlike the rest of this paper, contains limited experiments that are hard to generalize.

a. The stress granules e.g. already show vacuoles before RNA addition (which I assume is their equivalent of quench)? What is the explanation?

b. The reconstitution (in nuclear lysate etc.) details are not mentioned anywhere.

c. The authors argue that vacuole formation is induced by solidification? Can they report on data to confirm solidification? How do they know vacuoles are not pre-existing and just trapped from "escaping" by a gradually "gelling" interface? I would recommend toning down this section

d. It appears the solidification section proposes syneresis → can they highlight what the origin of composition or temperature change is in their DNA or stress granule data reported in ext fig 10B/C?’

Overall, in this reviewer's opinion, this section reads much more like a discussion section and would recommend implementing one of the following options:

(1) significantly expanding the stress-granule/DNA-antibody arguments to augment the current section and move the discussion-like exposition (starting lin 160/161) into the conclusions or as a separate discussion section.

(2) Alternatively, the authors could completely move the section into Discussion and continue to use limited but better clarified pointers to Stress Granule/DNA experiments.

(3) Finally, if the authors choose to remove Ext. Data 10 and treat the whole topic as an extended discussion, this would be largely fine as well.

Reviewer #2 (Remarks to the Author):

Very nice work. Please see my attached review (PDF).

This is a very nice, well-written and clear review of a topic of emerging importance and interest -I enjoyed reading this greatly. The authors do a good job of balancing complex ideas in an accessible way, while (implicitly) introducing many of the more important questions in the field. My comments below should be taken in the way they are intended; as friendly suggestions/ thoughts one might offer to a friend, as opposed to harsh criticisms.

Major comments:

- In section 1, the authors organize organizing principles by scale, yet this seems somewhat arbitrary and not rigorously correct. Vesicles can be 10s of nanometers across, and the nucleolus has historically been one of the largest structures in the cell, upending the 'membrane big, condensate mesoscale' narrative. I would suggest the authors move away from implying that size is a defining feature that is relevant for membrane-less vs. membrane-bound organelles; this doesn't seem to hold up to scrutiny and is possibly more confusing than helpful. It's of course fine to talk about organization at those scales, so the review does not need restructuring, but even at the microscale complexes can be well-defined and stoichiometric or highly disordered lacking structure (see Borgia et al. Nature 2018).
- *"Traditionally, protein assembly in biology has been viewed as a stepwise process that follows a delineated path, giving rise to structures such as a ribosome where each molecular species occupies a specific position in space."* - I worry that by describing these assemblies as 'stepwise' the authors have set an impossible gauntlet - do we know the order of the steps of assembly, and do we know if that order matters? This idea has shades of protein folding pathways vs. energy landscapes vs. foldons, ideas that really led to discontent and (in this reviewer's opinion) decades of pointless arguments in the protein folding community. I wonder if rather than thinking about the process of assembly, if the authors could define complexes as 'stoichiometric' vs. non-stoichiometric assemblies? I.e. traditionally biology has been focussed on defined 'stoichiometric' complexes, but condensates do not require a strict and unwavering stoichiometry. This is a looser and more agreeable definition than simply stating that a specific stepwise process exists, which may or may be demonstrably true and/or demonstrably important. The writing is basically there anyway, and while superficially the idea of 'step wise assembly' is nice, my concern is that when one starts thinking harder about what this means challenges begin to emerge,
- The authors seem to imply that condensates must form through phase separation. This is not correct - the original definition of condensates (Banani *et al.*, Cell 2017) states: *"Here, we propose a new name — biomolecular condensates — which emphasizes the two features common to all of the structures: their ability to concentrate molecules and that they comprise biological molecules, independent of all other characteristics"* - i.e. the term biomolecular condensates does not prescribe a mechanism of formation, not does it prescribe a material state of the associated assembly. The authors should make this point clear, and that this discussion is focussed on condensates formed through phase

separation.

- “Liquid-liquid phase separation necessitates that the conformational flexibility of the subunits is amenable to the high entropy of a liquid phase when compared to the solid phase. Therefore, a high degree of conformational freedom is an essential feature of polymers which readily undergo liquid-liquid phase-separation.” - I’m not sure this is true; one can run a simulation of simple Lennard Jones spheres and obtain conditions in which coexistence of a dense and dilute liquid phase emerges. Patchy colloids (where molecules have no conformational flexibility) readily undergo phase separation (see Espinosaa et al. PNAS 2020, although there is a large body of work in this area). Lysozyme undergoes liquid-liquid phase separation (Taratuta et al. JPC 1994). Whether conformational flexibility enables a saturation concentration to exist in a regime that is biologically accessible is a hypothesis that remains to be tested, but the idea that conformational flexibility is required is demonstrably not correct.

Minor comments:

- The second paragraph is a strange single sentence - feels like it’s missing some content? In general paragraphs shouldn’t be single sentences...
- “Notably, in recent years, IDRs **were** frequently observed” → “Notably, in recent years, IDRs **have been** frequently observed” (‘were’ implies an event that ended, ‘have been’ implies a continuing observation first made earlier in time; very minor suggestion but helps make clear that IDRs continue to mediate compartmentalization.
- When IDRs are first introduced there are no references - I might suggest the following three as good foundational references
 - Wright, P. E., & Dyson, H. J. (1999). Intrinsically unstructured proteins: re-assessing the protein structure-function paradigm. *Journal of Molecular Biology*, 293(2), 321–331.
 - Dunker, A. K., Brown, C. J., Lawson, J. D., Iakoucheva, L. M., & Obradović, Z. (2002). Intrinsic disorder and protein function. *Biochemistry*, 41(21), 6573–6582.
 - van der Lee, R., Buljan, M., Lang, B., Weatheritt, R. J., Daughdrill, G. W., Dunker, A. K., Fuxreiter, M., Gough, J., Gsponer, J., Jones, D. T., Kim, P. M., Kriwacki, R. W., Oldfield, C. J., Pappu, R. V., Tompa, P., Uversky, V. N., Wright, P. E., & Babu, M. M. (2014). Classification of intrinsically disordered regions and proteins. *Chemical Reviews*, 114(13), 6589–6631.
- When discussing condensate kinetics (page 4) it would be worth citing - an elegant dissection of condensate kinetics using SAXS (Martin et al, Nature Comms. 2021). It may also be worth discussing work from the Brangwynne lab where critical nucleation theory appears to offer a quantitative description of intracellular condensate kinetics.

- When discussing Flory Huggins the authors state “*However, the model is not complete enough to describe the phase separation of biopolymers, since the identity of each subunit is assumed to be the same*” - this is technically correct, but there’s some nuance here; many heteropolymeric IDRs can be cast as homopolymers, at which point Flory Huggins works remarkably well. It may be worth slightly expanding this - e.g. “However, Flory Huggins prescribes every subunit (e.g. amino acid) with the same chemical identity, a simplification that can be phenomenologically useful but is unable to capture the reality of chemically complicated polypeptides”
 - References to where this has worked well include
 - Nott, T. J., Petsalaki, E., Farber, P., Jervis, D., Fussner, E., Plochowietz, A., Craggs, T. D., Bazett-Jones, D. P., Pawson, T., Forman-Kay, J. D., & Baldwin, A. J. (2015). Phase transition of a disordered nuage protein generates environmentally responsive membraneless organelles. *Molecular Cell*, 57(5), 936–947.
 - Brady, J. P., Farber, P. J., Sekhar, A., Lin, Y.-H., Huang, R., Bah, A., Nott, T. J., Chan, H. S., Baldwin, A. J., Forman-Kay, J. D., & Kay, L. E. (2017). Structural and hydrodynamic properties of an intrinsically disordered region of a germ cell-specific protein on phase separation. *Proceedings of the National Academy of Sciences of the United States of America*, 114(39), E8194–E8203.

- It may be worth noting that the stickers-and-spacers model is not restricted to flexible polymers, but more broadly can be extended to describe biomolecules where the surface can be easily divided into sticker regions and spacer regions (see Choi et al *Ann. Rev. Biophys.* 2021)

- When discussing crowding agents, it would be prudent to mention the fact that *bona fide* inert crowders are generally a thing of theory, not of the real world; weak interactions with hyper-valent crowders may also drive phase separation through heterotypic attractive or repulsive interactions - see
 - Qian, D., Welsh, T. J., Erkamp, N. A., Qamar, S., Nixon-Abell, J., St George-Hyslop, P., Michaels, T. C. T., & Knowles, T. P. J. (2022). Tie-lines reveal interactions driving heteromolecular condensate formation. In *bioRxiv* (p. 2022.02.22.481401). <https://doi.org/10.1101/2022.02.22.481401>

- When discussing collapsed states of proteins (page 12) it may be good to mention this preprint which shows the same principles explored for disordered proteins can be applied to understand unfolded states of conventionally foldable proteins.
 - Ruff, K. M., Choi, Y. H., Cox, D., Ormsby, A. R., Myung, Y., Ascher, D. B., Radford, S. E., Pappu, R. V., & Hatters, D. M. (2021). Sequence grammar underlying unfolding and phase separation of globular proteins. In *bioRxiv* (p. 2021.08.20.457073). <https://doi.org/10.1101/2021.08.20.457073>

- When discussing the impact of the cellular environment - the idea that IDRs are inherently sensitive to changes in the local solution environment has been examined both in terms of the intrinsic physicochemical basis of sensing (Moses et al. JPCL 2020), in the context of pH sensing (Franzmann et al Science 2018, Gutierrez et al. eLife 2022) and in the context of temperature sensing (Riback et al. Cell 2017) and as general sensors (Meyer, COPB 2020). It might be good to include these citations here.

Reviewer #3 (Remarks to the Author):

In this manuscript, Erkamp and colleagues describe in vitro experiments with PolyA RNA + PEG showing that condensates adopt multiphase structures via a kinetically arrested phase transition. They show that cavities form at 20C and dissolve at 55C, in a reversible manner. By showing reduced cavity formation at lower cooling rates, they show that cavitation is a kinetic process rather than a thermodynamic one. In order to analyze the composition of fluid within the cavities, capillary velocity was evaluated by measuring the aspect ratio, width, and length of various cavity fusion events. Cavities were ruled to be similar in composition to the bulk dilute phase. Binodal and spinodal curves were calculated using multiple models, and the formation of cavities is represented by deviations from the binodal towards the spinodal, due to slow diffusion of PEG and PolyA. Finally, reconstituted stress granules show cavity fusion and enlargement over time, even without addition of RNA or temperature change, pointing to a more permanent gel-like behavior.

This work contributes to the effort of understanding biomolecular condensates, which is a very timely topic. The authors make strong arguments for the formation of cavities as an inherent, kinetically driven process in their in vitro system, which is interesting. However, the biological relevance of this observation is unclear, with the size, composition, the condensate environment and activity being far different from in vivo conditions. Because of this concern, I think the paper is better suited for a specialized journal and I cannot recommend publication in Nature Communications.

Major Concerns:

1. As mentioned above, the biological relevance of the presented observations is unclear: The authors report that cavities only form in large condensates not in the small ones. For example, Fig 2 shows that at a condensate radius of 14 μm , cavities are never formed. While this size is small relative to the other condensates studied in this work, it is huge compared to the condensates found inside the cell. Given that most cellular biomolecular condensates are one micron or smaller in size, it suggests that such cavitation could not occur in condensates inside the cells.

Further, the observed diffusion rate of PEG is $\sim 12 \mu\text{m}^2/\text{s}$, much faster than the $\sim 1 \mu\text{m}^2/\text{s}$ measured for in vivo condensates, which could explain the lack of cavitation at small sizes. However, Extended Data Fig 9 shows a diffusion rate of $2.5 \mu\text{m}^2/\text{s}$ for PolyA, much closer to true biological regimes, which again raises question of biological relevance. A discussion is needed of why polyA condensates do not form cavities at less than a $14 \mu\text{m}$ radius. The effect of size scaling and the translation of the observations to small biologically relevant sizes should be discussed.

2. In addition, the smaller condensates might be dominated by surface effects, which are negligible for large in vitro condensates. This should be considered, as this can cause different types of behavior at small sizes, which are biologically relevant.

3. The authors observe that the cavities have similar/same composition as the surrounding solution of the condensates. This is in contrast to in vivo condensates, whose subcompartments are biopolymer-poor, but contain yet another biomolecule that makes up the cavity, in other words they are of different composition than the surrounding solution. If indeed these cavities form initially simply with the dilute phase, then the necessary biomolecules would have to pass through the dense phase into the newly formed cavities. This would disrupt the kinetics of the condensate and possibly terminate the kinetically arrested phase transition. So, it is not immediately clear, how to apply the observations of this paper to in vivo systems.

4. The authors address the *in vivo* relevance of Extended Data Figure 10, but the data presented is quite minimal. Although the Materials and methods section describes in detail the complex procedures for preparation of the reconstituted stress granules and antibody-DNA condensates, there is no analysis presented, only a presence of cavities is shown. The kinetics of these cavities is not analyzed in a quantitative way, which would support the claims in this work. Additionally, the size regimes are still an order of magnitude above those in *in vivo* biomolecular condensates. In fact in Extended Data Figure 10a, it is clear that the smallest droplets do not form cavities. Since these are the most biologically relevant cavities, some discussion should be focused on them and corresponding kinetics analysis of this data should be added.

5. Another difference between the presented *in vitro* condensates and *in vivo* condensates is the presence of activity in the latter. This is only very briefly mentioned in the manuscript, but is a central point of living systems and hence should be adequately addressed. Could authors elaborate how would activity impact their presented picture?

6. In Extended Data Fig 2, both PolyA and PEG signals are shown. In Fig 1-3, there is no information provided if the signal shown is polyA or PEG. Based on the shape of cavities shown in Extended Data Fig 2, it would appear that the cavities are more spherical in PEG, which is very similar to the signal shown in Fig 1-3 and thus suggest we are looking at PEG signal. Yet, the phase diagram is based on the PolyA concentration. This should be clarified. It would be helpful to see some more examples of PolyA cavities. They seem less spherical than the corresponding PEG cavities, is there a critical size where polyA doesn't form cavities but PEG does? The work would be improved by a discussion on the differences in cavity structure in the two signals. Figures should also be clearly labeled to indicate whether it is PolyA or PEG being imaged.

7. In addition to different appearance of cavities in polyA vs PEG, their diffusion rates are also quite different ($2.5 \text{ } \mu\text{m}^2/\text{s}$ vs $12 \text{ } \mu\text{m}^2/\text{s}$). The authors use pictures of PEG and polyA interchangeably, implying that wherever there is a PEG cavity there is also polyA cavity. However with such different diffusion rates, the kinetics of PEG and polyA may affect the cavity formation. To address this, both signals in Fig 1-3 and in Extended Data Fig 9 should be shown.

8. Fig 3 discusses the fusion rate of cavities within condensates, but the size of these cavities is not discussed. Do cavities fuse at constant rates in all scenarios, or are they dependent on condensate size or individual cavity size?

Minor Issues:

1. In Figure 2, the temperatures in the subscript of radius in the top row are too small to be easily legible.

2. Lines 57, 340: form is misspelled as "from"

Reviewer #4 (Remarks to the Author):

In this work, the authors present results from measurements of synthetic condensates and demonstrate that vacuole (or cavity) formation is a dynamically controlled phenomenon. The results are presented for a ternary mixture comprising solvent, PEG, and poly-rA. The

results are very interesting and likely to be of broad interest and relevance.

While a series of interesting findings are presented, what is lacking is two-fold. The phrasing (see specific comments below) of various aspects of the study and the lack of distinction between spatially organized multiphase structures in multicomponent systems vs. vacuole formation (observed in yeast nucleoli) is not clearly made. This is relevant because the extant literature clearly lays out the distinction between multiphase structures coming about from thermodynamic control vs. vacuolized structures forming via dynamical control. Second, a very interesting body of data is presented. However, a robust theoretical or phenomenological computational model is what is needed to tie things up. This is missing. If these changes can be made, then this would make for a very interesting and timely contribution that deserves to be published in Nature Communications. The specific comments are categorized as major and minor.

Specific major comments

1) Distinction between multiphase, spatially organized structures vs. structures with holes i.e., vacuoles. The introductory narrative takes a bit of parsing because it is not clear if the authors are referring to multilayered structures such as nucleoli or if the focus is on observations (less in biology and more in the synthetic literature) of vacuoles that appear to create the impression of coexistence of filled and hollow phases. Please consider rewriting the introduction to clarify the problem of interest. Once this is done, and if one does introduce the topic of true core-shell architectures, then please note that there is a well honed thermodynamic basis for the appearance of such structures. This was first put forth in the work of Feric et al., ([http://www.cell.com/cell/fulltext/S0092-8674\(16\)30492-5](http://www.cell.com/cell/fulltext/S0092-8674(16)30492-5)). Please note that even the simplest ternary mixtures of polymers can have core-shell and vacuolar structures. These have been explained on the basis of sequence-specific immiscibility. In addition to the work of Feric et al., and Fei et al., there is the work of Harmon et al., which shows just how one can get spatially organized droplets out of differential solvation effects. Please see: <https://iopscience.iop.org/article/10.1088/1367-2630/aab8d9>. Likewise, Simon et al., show this using ELPs of different hydrophobicities <https://www.nature.com/articles/nchem.2715>. The key message is that there is a solid thermodynamic framework for spatially organized droplets. However, the presence of vacuoles and the emergence of vesicles is less clear in terms of the interplay between thermodynamic vs. dynamical control. The work of Boeynaems et al., (which showed a thermodynamic basis for the spontaneous emergence of spatially organized structures) does demonstrate how dynamical arrest can interfere with phase separation, as first illustrated by Sciortino et al. Please see: <https://www.pnas.org/content/early/2019/03/28/1821038116> and references therein. Further, in a recent study, Seim et al., have demonstrated that "off-pathway" oligomerization via homotypic interactions can impact the protein density and even the rearrangement dynamics of protein-RNA condensates that form mainly via heterotypic interactions. Please see: <https://www.pnas.org/doi/10.1073/pnas.2120799119>. The key thrust of the point being made is that spatially organized condensates can arise, and possibly do arise, from purely thermodynamic considerations. Vacuole formation continues to be a puzzle, and it is satisfying to see a more intuitive dynamical argument being put forth here. These points, with appropriate nods to the literature, could be better clarified in the narrative, both in the introduction and in the connections made between the observations and the literature. As currently crafted, some of the verbiage is stilted and this is likely to engender a lot of confusion. In this context, it is perhaps a stretch to assert that "While this class of structure is thus commonly encountered in nature...". It is not clear that such structures are common in nature.

2) At the risk of seeming like a broken record, please clarify that the structure being

investigated is that of a condensate with holes i.e., a condensate undergoing vacuolization. This is super relevant because as the authors clearly note, they are studying the formation of "cavities, liquids poor in PolyA and PEG inside the condensate". Indeed, even the title is misleading. It should be something like: Condensates with holes form via dynamical arrest or something like this because almost every reader will jump to the conclusion that facsimiles of nucleoli arise due to dynamical arrest, and this is clearly not the case.

3) The relevance of the work of Simon et al. (see above) cannot be overemphasized.

4) This particular statement "If cavity formation is a kinetic process we would expect cavities to form at fast composition changes, whereas cavities would be formed at slow composition changes if it is a thermodynamic process" is concerning because the standard definition of thermodynamic control is of reversibility and reproducibility no matter the starting point. It appears the statement is intended to distinguish between quenched disorder vs. the lack of such disorder upon annealing. This interpretation is consistent with the data in Figure 2, which are really neat. Please consider rewording.

5) To give credit where it is due, Banerjee et al., did make the point that prior to dissolution, the peptide-RNA condensates they studied go through vacuole formation. Essentially they viewed this as nucleation of the dilute phase within the condensate. I refer here to reference 12, which could be better cited and integrated in with the current results.

6) Please elaborate on the details of the "three classes of fusion events".

7) This sentence is difficult to parse because the meaning of it is not clear: "Since the viscosity entering the capillary velocity equation is always that of the dense condensate phase, any differences show differences in the surface tensions between the different interfaces." Please reword and clarify.

8) The current system is a ternary mixture comprising the solvent, PEG and poly-rA. If the concentrations of the macromolecules are variables, and so is the temperature, then the phase boundary is not longer a binodal. This verbiage is problematic because it is misleading. It appears that the authors are describing a mechanism where the solvent activity is fixed as is the concentration of PEG. Then the variables are the concentration of poly-rA and temperature. Is this the case? It appears to be the case based on Figure 4A. If so, this should be specified very clearly. However, it is then incumbent to specify that for fixed amounts of PEG, the concentration titrations of poly-rA cause significant sub- and super-stoichiometric regions to be explored along the phase diagram. Inasmuch as this is the case, and given that the ratios of molecules that combine to drive phase separation are viewed as the relevant order parameters for vacuolization and / or reentrant phase behavior, the stoichiometric ratios and their connection to cavity / vacuole number and size become super important and relevant.

9) Overall, the lack of a theoretical framework or a computational model that explains the data leave one wanting more. Given that cavity formation can be observed in silico, one can then investigate the effects of a deep or shallow quench and timescales in dynamics simulations or move sets in MC simulations. Examples of this abound in the soft matter and biological phase separation literature. Right now, the MS ends with a set of observations, a somewhat arbitrary classification of viscoelastic vs. liquid-like materials (should this not be dependent on the dynamical moduli), and a proposal that reads a bit like an assertion. Having a model that incorporates the measured parameters or a simulation that captures the phenomenology would be considerably more satisfying.

Specific minor comments

**1) Please note that paraspeckles are not condensates in the conventional sense. They are micelles that undergo sphere to rod transitions. Please see:
<https://www.embopress.org/doi/full/10.15252/emj.2020107270>.**

2) Given the focus on spatial inhomogeneities within condensates, the prefix of LL in LLPS is misleading. The term serves as a straw-man, which almost all reasonable enthusiasts and data will take down. Please consider deleting all mention of LL and / or liquid-liquid, at least in the context of biological phase separation. For the synthetic system here, it probably makes sense. Also, when discussing liquids, please specify what types of liquids one should conjure up as a models.

3) As a matter of taste "binodal curve" should be just binodal.

Reviewer #1:

In this study by Erkamp, Sneideris et al., the authors propose that kinetic arrest could contribute to formation of condensates with cavities, which they term “multiphase” condensates. They develop a model system of RNA and PEG phase separation and tunably induce cavity/vacuole formation through temperature sweeps. Through a clean assay modulating the importance of kinetics vs thermodynamics by changing the rate of temperature change, they identify features that contribute to cavity stability and size. They identify that these cavities resemble the dilute phase and undergo merging and growth. Overall, they propose that the quick quenches move the system closer to the spinodal, which due to the slower diffusion of biomolecules in the dense phase, leads to nucleation of vacuoles within the dense phase. While they develop their assays with temperature quenches, they hypothesize the generality with different quench/annealing parameters that control the importance of kinetics i.e. including pH, salt, and composition.

Overall, the paper is compelling and an interesting study. The development of their assay is well-motivated and by large, the conclusions are well-supported by the experiments. I outline a few concerns/changes - intended to clarify presentation, improve quantification and/or reanalyses of existing data-sets to strengthen existing claims, and re-evaluate certain experiments. If addressed, I believe the manuscript will be improved and would recommend publication.

We thank the reviewer for their assessment of the manuscript and their fantastic feedback.

Concerns:

1. Terminology: The authors use the term “multiphase” condensate broadly to discuss their specific example - of dilute phase nucleation arising from kinetic arrest in the dense phase. I am puzzled by this choice of wording as one is led to believe there are more than 2 phases existing in systems - indeed, this is not the case. Rather, rapid shifts in composition lead to heterogeneous nucleation within dense phases, a phenomenon known before in different contexts. As the authors motivate themselves, the literature references include 2 distinct phenomenology → examples like the nucleoli or paraspeckles are multiphase in the sense that their structure is characterized by more than one order-parameter (depending on specific combination of compositions in each phase) and a secondary case with vacuoles or cavities (especially for the system motivated here - a single equilibrium order parameter suffices). I believe that this essential issue needs to be adequately addressed:

a. Considering rewording description of their system to “vacuole” or “cavity” or “heterogeneous” or an equivalent example the authors may come-up with to replace multiphase.

We thank the reviewer for drawing attention to the terminology used throughout the manuscript. Indeed, the term multiphase implies more than 2 phases, while one of the most important messages of the paper is that the structure of the condensates is formed by the dilute phase being trapped inside of the dense phase, rather than a third phase. We have changed “multiphase” throughout the manuscript, including in the title, to “double emulsion” or “spatially non-uniform”, as this more accurately describes the structure. In line with this comment, we have also changed “cavity” to “trapped (dilute phase) droplet” and “kinetically arrested phase separation” to “dynamically arrested phase separation”.

b. In their introduction, the authors directly jump from “multiphase subcompartments” (which are not biopolymer poor, often different in relative concentrations and distinct from nucleoplasms) to “vacuoles” and “core-shell” architectures. I would recommend clarifying that the “specific” examples listed constitute cases where the internal subcompartments of vacuoles or cavities are biopolymer depleted.

This is a good suggestion which will further help with clarifying how multiphase structures are different from the double emulsion systems in literature and our manuscript. We have changed the wording in the introduction to highlight this contrast (line 36).

2. Quantification of cavity sizes and number: While the visual image for cavities changing as a function of temperature are appealing, statements that comment on cavities growing smaller or larger and reversibly would be better served by quantification. To this end, I recommend incorporating analyses/text that directly addresses:

a. Incorporating a condensed data-plot similar to Ext Data Fig 6 in Figure 2 that shows the variation in cavity number, cavity size etc. with different condensate sizes and cooling rates.

We thank the reviewer for their suggestion. The data shown in Ext. Data Fig. 6 shows the number of trapped droplets for each condensate size and each cooling rate. We have emphasized that this data can be found here in lines 88-89. We have also changed the legend of Ext. Data Fig. 6 to state the radius of the droplets, rather than the diameter, matching with Fig. 2.

b. One expects the nucleation rate to scale non-linearly with super-saturation. Is the cooling rate a proxy for super-saturation? If so, is Ext. Data 6 consistent with this? Can the authors also specify what time-point the data is reported for number?

This is a particularly good observation from the reviewer. Indeed, cooling rate causes an effect similar to super-saturation, except that in super-saturation the concentration in the solution exceeds the solubility, while in our case, the concentration is lower than the binodal concentration (Figure 5a). Similar to super-saturation, nucleation occurs more often the further from equilibrium/solubility the system is (Ext. Data Fig. 6a). Notably, the reviewer points out that we expect the nucleation rate to scale non-linearly with getting closer to the spinodal. We do not observe this specific trend here, since the cooling rate does not translate 1:1 to distance from the spinodal. The reviewer also asks to include the time after cooling until the measurement is taken place. This is 1 minute and is added this at line 407.

c. Is there a reproducibility in the number or size of cavities? For e.g. do the numbers average for the same-condensate across multiple, well-separated temperature cycles? How about for 2 condensates of similar sizes?

Including such analyses more broadly, including for Ext Data Fig 3. should be fairly straightforward and will improve interpretation.

Great suggestion. We have quantified the number of dilute phase droplets at 20 °C in figure 1 and Extended Data Fig. 3. The condensate has a similar size before and after each cycle, thus making this data suitable to check the reproducibility of the number of trapped liquids. We find that the number of trapped droplets in this condensate is: 106 (left Fig. 1), 101 (middle Fig. 1), 93 (right Fig.1), 107 (left,

Ext. Data Fig. 6), 96 (middle, Ext. Data Fig. 6), 98 (right, Ext. Data Fig. 6). Thus, the number of trapped droplets in a condensate of this size when cooling 20 °C/min is 100 ± 5.6 . Given the low standard deviation and the number of cycles over which we can form similar amounts of cavities, this behaviour is well reproducible. In the manuscript, we have highlighted this reproducibility and added our result of “ 100 ± 5.6 ” trapped liquids (line 57-59).

3. Cavity dynamics: Motivated by (2), it would be interesting to report on dynamics of condensate sizes. For e.g., in the data reported in Fig. 3 → does the cavity formation have any resemblance to classical nucleation/growth w.r.t scaling laws or distributions in size?

I note here that no new analyses is strictly required and reporting the variations as a function of time would be sufficient, although the authors may choose to add-in more along the lines suggested outlined above.

We thank the reviewer for this truly fantastic comment. We have investigated whether the behaviour of the trapped liquids follows classical nucleation and growth scaling laws using the data from Figure 3A and supporting video 1, in which the trapped liquids fuse and grow. Thus, as the reviewer recommends, we can report the number and average radius as a function of time in Fig 3b and c.

The number of trapped droplets decreases as they fuse with each other or fuse with the surrounding dilute phase via $D + D \rightarrow D$ and $D + P \rightarrow P$ respectively.

This reaction gives us: $\frac{d[D]}{dt} = -k [D] ([D] + [P])$

$$\frac{d[D]}{[D]([D]+[P])} = -k dt$$

$$d[D] \left(\frac{1}{[D]} - \frac{1}{[D]+[P]} \right) \frac{1}{[P]} = -k dt$$

We integrate this to give: $\ln([D]) - \ln([D] + [P]) = -[P]kt + C$

$$\frac{[D]}{[D]+[P]} = C e^{-[P]kt}$$

$$[D] = C e^{-[P]kt} ([D] + [P])$$

$$[D] = \frac{[P]C e^{-[P]kt}}{1 - C e^{-[P]kt}} = \frac{[P] e^{-[P]k(t+t_0)}}{1 - e^{-[P]k(t+t_0)}} = \frac{\alpha e^{-\alpha\beta(t+\gamma)}}{1 - e^{-\alpha\beta(t+\gamma)}}$$

Thus, our number of droplets $N(t)$ should depend on t via the formula $N(t) = \frac{\alpha e^{-\alpha\beta(t+\gamma)}}{1 - e^{-\alpha\beta(t+\gamma)}}$

Using this formula, we obtain a very good fit for our data (Fig. 3b) with $N(t) = \frac{29 e^{-29 \frac{1}{13436}(t+104)}}{1 - e^{-29 \frac{1}{13436}(t+104)}}$.

Additionally, we have determined how the radius of the trapped droplets should depend on time (Fig. 3c). Lifshitz-Slyozov-Wagner (LSW) theory has demonstrated that when small droplets fuse to larger droplets, the radius scales with $R(t) \propto t^{\frac{1}{3}}$ (section 2.5 and 2.6 in [1]).

When fitting our data with an equation of the shape $R(t) = \gamma(t + \delta)^{\frac{1}{3}}$, we obtain a good fit with $R(t) = 0.5486(t + 38.58)^{\frac{1}{3}}$.

Thus, both the number of droplets and the radius of the average droplet over time fit well with the expected scaling laws. The graphs have been added to Fig. 3, the derivation above has been added as Supplementary text 1 and the main text has been changed to discuss these two new subfigures (lines 116-120).

[1] Bray, A. J. (1994). Theory of phase-ordering kinetics. *Advances in Physics*, 43(3), 357–459

4. Figure 4A, while motivated in good spirit, is a little confusing with binodals/spinodals since T is used as the experimental knob throughout this study. I recommend either changing or including a version specifically with temperature (like the Ext. Data Fig)

Again, an excellent point from the reviewer. We have changed previously figure 4A, now figure 5b, to have temperature on the y-axis. Additionally, we have included the “dilute side” / “left side” of the boundary to make it easier for the reader to orientate themselves.

5. As written the “slow diffusion ...” section is a little dense. Can the authors, by text or otherwise, clarify the logic behind the diffusion limited argument i.e to show that concentration increases require diffusion across “length-scales” measured by $\langle x \rangle$ and thus droplets larger than that compensate for this change through nucleation of dilute phases? If the reviewer misunderstood the

argument, clarity by rewriting text could only help.

We thank the reviewer for their comment. Indeed, condensates larger than a critical size cannot equilibrate to the binodal sufficiently due to slow diffusion. Thus, the composition change is achieved by nucleating the dilute phase inside of the condensate. The reviewer understood that argument correctly. Notably, this is an important message of the paper and should be clearly explained in this section. Thus, we have made changes to lines 225 – 237.

6. The last section on kinetic arrest in diverse systems is confusing and unlike the rest of this paper, contains limited experiments that are hard to generalize.

We have addressed this concern by performing additional experiments with the reconstituted stress granules and quantified this (Fig. 4 and Ext. Data Fig. 7)

a. The stress granules e.g. already show vacuoles before RNA addition (which I assume is their equivalent of quench)? What is the explanation?

This possibly happened directly after formation of the condensates when we placed the sample onto the microscope and imaging stage, which was 20°C rather than the temperature of the room. To prevent this, we have removed this data and have added new experiments where the reconstituted stress granules do not contain any trapped droplets yet, since they were prepared on this stage and thus without temperature changes before the start of the experiment (Fig. 4 and Ext. Data Fig. 7).

b. The reconstitution (in nuclear lysate etc.) details are not mentioned anywhere.

They are stated in the supplementary information, instead of the main text. The procedure for reconstituting the samples can be found in the Supplementary Information lines 214 - 218 and the material preparation is outlined in lines 259 - 304.

c. The authors argue that vacuole formation is induced by solidification? Can they report on data to confirm solidification? How do they know vacuoles are not pre-existing and just trapped from “escaping” by a gradually “gelling” interface? I would recommend toning down this section

Because it was difficult to quantify trapped droplet formation because of “solidification” we have indeed removed this data and focussed on quantifying the formation of trapped droplets after RNA addition.

d. It appears the solidification section proposes syneresis → can they highlight what the origin of composition or temperature change is in their DNA or stress granule data reported in ext fig 10B/C?

Overall, in this reviewer's opinion, this section reads much more like a discussion section and would recommend implementing one of the following options:

(1) significantly expanding the stress-granule/DNA-antibody arguments to augment the current section and move the discussion-like exposition (starting lin 160/161) into the conclusions or as a separate discussion section.

(2) Alternatively, the authors could completely move the section into Discussion and continue to use limited but better clarified pointers to Stress Granule/DNA experiments.

(3) Finally, if the authors choose to remove Ext. Data 10 and treat the whole topic as an extended discussion, this would be largely fine as well.

We thank the reviewer for this comment. We have decided to significantly expand the first section of the original Ext. Data Fig. 10, where RNA is added to the stress granules. By preparing the condensates at constant temperature, we have prevented that the condensates contain trapped droplets before the addition of PolyA RNA and dye. Then, we have added the PolyA RNA to the solution, the condensate changes composition as it takes up some of this new material and thus forms trapped dilute droplets, poor in both G3BP1 protein and nucleic acids, shown by the newly added dye (figure 4a and Ext. data fig 7). We have quantified the number of trapped droplets before and after the addition of PolyA and dye as a function of the radius of the condensate in figure 4b. This clearly shows that a double-emulsion structure can be obtained by composition change, but also that the number of trapped droplets is highly dependent on the size of the condensate, as was also observed in the PolyA-PEG system (lines 147 – 155).

We thank the reviewer again for taking the time to give feedback on our manuscript. Their comments were fantastic!

Reviewer 2:

This is an interesting manuscript that quantitatively addresses an important issue surrounding the basis for vacuole or cavity formation in condensates. Conceptually I think the work is sound, but I have a number of issues which are outlined below.

We thank the reviewer of taking the time out of their schedule to give feedback which has helped with improving our manuscript. Below, we respond in a point-by-point fashion to all the comments.

Big picture questions

1. This feels like a cruel question but, do the authors anticipate a polyA/PEG coacervate system to generalize to other condensates, and if so why? I'm not necessarily arguing it doesn't, but the motivation or using this model system was weak, and the additional systems investigated in the final selection didn't really add much, in my opinion, to sway these concerns.

We thank the reviewer for raising this question. We have performed further experiments on the reconstituted stress granule system (Fig. 4 and Ext. data Fig. 7) to show that this mechanism also applies to this system, despite the condensate being made of a different material and the composition change being induced by RNA addition, instead of temperature changes. We have also modelled the

effect of fast and slow composition changes in silico (Fig. 5a, supplementary video's 2, 3, 4 and 5 and supplementary text 2) to provide further evidence that this mechanism allows phase separating systems to form a double-emulsion structure upon fast composition changes, independently of the compounds making that system up or the source of the composition change. Indeed, PolyA-PEG condensates are used as a minimalistic model system to obtain mechanistic insights into the origin of this structure (lines 50 – 52). We think the additional data showing that reconstituted stress granules and condensates in general (in silico) can achieve this structure will help with the generalisation of this mechanism from our model system to different systems.

2. The manuscript is incredibly short, and would benefit from a substantial re-write to expand the content, bring more quantification and perhaps figures from the SI into the main text, and really walk the reader through. Specifically, I would want to see the specific open being addressed clearly articulated. Right now it sort of reads like a list of things that were done (and as best I can tell done well!) without really motivating why they were done.

We appreciate that the reviewer is interested in the expansion of the work. We have expanded and further quantified the figures (Figure 3, 4, 5, Supplementary text 1) and added a model (Supplementary text 2). We have also expanded the main text to more thoroughly walk the reader through the results (lines 112-250).

3. This is a strange question but - who are the author's intended audience? The analysis of the data felt largely phenomenological and descriptive; and the Flory-Huggins description while nice (and, for the record, I think 100% correct) didn't really connect back to the experiments quantitatively, so I'm not sure the theory folks would be sated. I'm also not convinced biologists will really care too much about this model system, so, am left wondering who the intended audience is.

With this manuscript, we aim to address a broad audience of researchers, both experimentalists and theoreticians, studying biomolecular condensates. We aim to show them that condensates with complex internal structure can be obtained via kinetic, rather than purely thermodynamic parameters. Most of the paper shows this message using a model system of PolyA-PEG condensates, which helps us to elucidate the mechanism at hand. We have added the information on stress granules on top of the referenced literature, to show this structure can form via the presented mechanism in a more biologically relevant system and have used an in silico model to show evidence that this is a general property of these systems. The Flory-Huggins description has helped us to construct the picture in figure 5b, which summarizes our findings. Given the fact that the formation of this structure was observed in multiple in vivo and in vitro condensate systems, we think that this work which outlines how this structure can be formed will be of interest to researchers working with them. We also believe this work is relevant for researchers studying condensates in general, as it shows that kinetic factors, which in studies often less highlighted than thermodynamic factors, are very important for understanding condensates and their structure.

4. In my opinion, demonstrating that vacuole:condensate surface tensions are indistinguishable from condensate:bulk surface tensions is the most interesting result, and I would encourage the authors to make this clear.

We thank the reviewer for this positive comment. We have made changes in lines 122 – 144 to emphasize and clarify this result.

Major comments

1. The fact polyA and PEG for co-condensates is sort of glossed over, but probably should be expanded upon a bit in the introduction; what's the underlying physical chemistry that drives this (with refs)? I realize this is not the point of the paper at all, but I think it's important to orient

We thank the reviewer for this comment. PolyA has been known to phase separate, at sufficient salt concentration and suitable pH, as far as we know since 1967 [1]. In this paper, phase diagrams of PolyA against temperature and salt concentration are provided in figures 1, 2 and 3 [1]. More recently, Roy Parker's group has done some great work on RNA self-assembly, showing a phase diagram of yeast RNA at various amounts of PEG and salt, as well as the phase separation of PolyA, PolyC and PolyU, in figure 2 of [2]. This paper, also discusses the interactions which can cause PolyA to phase separate in the presence of PEG, namely trans-RNA-RNA interactions, not Watson-Crick base pairing. We have added this second paper as reference 27 when the model system is introduced (line 52).

[1] Eisenberg, H. and Felsenfeld, G.. Studies of the Temperature-dependent Conformation and Phase Separation of Polyriboadenylic Acid Solutions at Neutral pH. *J. Mol. Biol.* (1967) 30, 1737.

[2] van Treeck, B. et al.. RNA self-assembly contributes to stress granule formation and defining the stress granule transcriptome. *PNAS*, 2018, 115 (11) 2734-2739.

2. The authors discuss polyA RNA and PEG but don't upfront give us information on the molecular weight/degree of polymerization for either. While it's common to give RNA and PEG molecular weights, I do think providing the degree of polymerization as well would help calibrate the audience as to how large these molecules are compared to (say) proteins. This should be specified and defined when these species are introduced. Similarly, if PEG is required for polyA condensate formation, it would be useful to see a phase diagram with [PEG] on one axis and [polyA] on the other, in equivalent units (ideally mass concentration, as polar concentration here could be misleading).

We have added both the molecular weight and the degree of polymerization of PolyA and PEG to the main text (lines 51-52). As, mentioned in the reply to the previous comment, various phase diagrams showing the behaviour of PolyA in the presence of PEG, salt at different temperatures have already been published. In our study, a phase diagram where the composition change as a function of temperature is shown is very relevant. We have changed the phase diagram (previously figure 4A, now figure 5b) to show this phase diagram (data shown in Ext. Data Fig. 1a) and how deviating from the binodal leads to trapped droplets in the condensate.

3. It would be useful if – rather than only showing pictures – figures 1 and 2 could include quantified the phenomenon they are reporting on. In particular reproducibility here (i.e. error bars) would I think be important to assess, especially because my intuition says the variance in number/size of vacuoles may correlate with surface tension given the stochastic nature of vacuole formation.

We thank the reviewer for their comment. We have quantified the number of dilute phase droplets at 20 °C in figure 1 and Extended Data Fig. 3. We find that the number of trapped droplets in this condensate is: 106 (left Fig. 1), 101 (middle Fig. 1), 93 (right Fig.1), 107 (left, Ext. Data Fig. 6), 96 (middle, Ext. Data Fig. 6), 98 (right, Ext. Data Fig. 6). Thus, the number of trapped droplets in a condensate of this size when cooling 20 °C/min is 100 ± 5.6 . Given the low standard deviation and the number of cycles over which we can form similar amounts of cavities, this behaviour is well reproducible. In the manuscript, we have highlighted this reproducibility and added our result of “ 100 ± 5.6 ” trapped liquids (line 57-59).

Figure 2 is also quantified in Ext. Data Fig. 6a.

4. The writing is somewhat stilted, a little odd, and at times not quite grammatically correct. It is mostly still quite readable, but it might be helpful to solicit feedback on the writing from those outside of the field, both in terms of clarity of topic and in terms of continuity of prose (e.g. “Over time, the number of cavities decreases via these two mechanisms, as they form less, but larger cavities or “escape” to the outer dilute liquid, decreasing surface energy”). I also found the use ‘multiphase’ here to be a bit confusing, given what we appear to be examining here has largely been referred to as vacuoles. Strictly speaking, if the vacuoles are simply dilute phase, then these are by definition not multiphase!

The reviewer brings up multiple important points here. Firstly, we agree that the term “multiphase”, is confusing here, since multiphase implies the presence of more than 2 phases, while one of the main points of the paper is that it is the dilute phase trapped inside of the condensate, rather than a third phase. To more accurately describe this system, we have changed the term “multiphase” throughout the paper, including the title, to “double(-)emulsion(s)” or “spatially non-homogeneous”. In line with this comment, we have also changed “cavity” to “trapped droplet”.

Additionally, the reviewer finds that the writing of the paper can be improved, for which we have made significant changes throughout the manuscript. They specifically mention the sentence “Over time, the number of cavities decreases via these two mechanisms, as they form less, but larger cavities or “escape” to the outer dilute liquid, decreasing surface energy”, which has been edited for clarity as well (line 135-136).

5. There is a lack of error bars in many of the figures (Fig. S1, S3, S6, Fig. 3). These are really required for us to be able to interpret these data.

We appreciate this comment and have added these in Ext. Data Fig. 1 (error bars added and information in the description line 366 and 372), Fig.1 and Ext. Data Fig. 3 (as discussed in Major comment 3 of this reviewer, lines 57- 58) and Fig. 3 (we have quantified the number and radius of the trapped droplets (Fig. 3b and c) and fitted these with the expected functions (Supplementary text 1)).

Given that we do not have control over the exact sizes of the condensates that we make, it would be difficult for us to repeat the exact experiments shown in Ext. Data Fig 6A, where condensates with radii of 14.3, 23.7 and 37.3 μm were used, to obtain an error. Luckily, these experiments are very similar to those of Fig. 1 and Ext. Data Fig. 3, except that the condensates were different sizes. For these experiments, we find a low error on the number of trapped droplets that is formed.

6. The fits in 3b and 3c are pretty questionable (3b @ 35 degrees uses just 3 points, and 3C @ 35 degrees almost all of the determination of the fit from the lone point associated with the one small droplet). I’d encourage the authors to obtain more data points and bound the error on the fit via an appropriate error propagation. I realize this is probably time-consuming, but I think a few more points are probably necessary.

We thank the reviewer for this comment, which refers to figure 3b and 3c, or current figure 3d and 3e in which each datapoint corresponds to 1 fusion event. As the reviewer points out, the number of datapoints at some of the conditions is limited, indeed because obtaining high time-resolution video's of approximately similarly sized liquids fusing without them significantly moving in the z-direction is very difficult. Luckily, all of the fits have to go through 0,0 because droplets that have a diameter of 0 μm will fuse in 0 seconds. This gives us an "anchor" for fitting our line and greatly increases the confidence we have in the angle we obtain. For this reason, we feel confident in the determined angles, despite not being able to obtain as many datapoints as this reviewer would consider ideal. We have added this explanation in Supplementary Information line 249-250.

7. I found the fact that the axes for fig 3. are different to be quite confusing. If the objective here is to show that inverse capillary velocities are similar at similar temperatures, you're actually making it harder for the reader to see that but messing around with the ymax. I'd just show all between 0 and 30 tau(s) and condensate diameter from 0 to 70. Honestly, if you're objective is for us to see how similar inverse capillary velocities are for equivalent temperatures regardless of where the aspect ratio relaxation is coming from, show this as a figure of the three lines at 25 degrees from the 3 different sources on the same axes; don't make me integrate across 3 different panels on different axes!

The reviewer offers 2 alternatives for displaying the data in the original Figure 3 b, c and d to better get across the conclusion from this data. The first option the reviewer gives is to have the same y max (and xmax) to more easily compare the slopes of the fitted lines:

The second option looks as follows:

We prefer the first option, as it does a good job at showing that the slopes at the same temperature are very similar and also contains the data at other temperatures. Thus, we have added the first option to figure 3 (d, e and f).

8. Re: diffusion rates “ we found that diffusion through these dense condensates is indeed slow “ - slow compared to what? We need a reference point to understand the meaning of ‘slow’.

We agree, slow would have to be compared to something. Notably, the point we try to make in this part of the text is not per se that it is “slow”, but similar to the fit from Figure 5c. Thus, we have focussed on this similarity and have removed the comment.

9. To paraphrase, the authors state “To determine if diffusion of biopolymers in the condensate could be a limiting factor, we measured diffusion. Diffusion was slow, and comparable to values measured previously. Therefore diffusion is the limiting factor” This doesn’t really follow, logically; I think if the authors wish to conclude that slow diffusion underlies this behavior they need to explicitly test this somehow. Otherwise, the paragraph maybe should focus on the absence of an elastic component in vacuole formation.

We have rephrased this to state more clearly that diffusion alone explains the location of the boundary between forming trapped droplets (line 234-235).

10. The final section feels like a strange smorgasbord of observations without any real integration into the rest of the paper. It’s possible it could contribute to the content, but, I’m not convinced it really adds anything as of yet.

This is a very helpful comment from the reviewer. To improve this, we have performed additional experiments on the reconstituted stress granules, which is the current figure 4 and Ext. Data Fig. 7. We have quantified the trapped droplet formation in these condensates to show that, similar to the PolyA-PEG system, larger condensates contain more trapped droplets. The text describing these observations (line 145 -164) is now approximately half-way through the main text at a logical point in the story, rather than added at the end of the paper. The main text now ends on the description of the mechanism, dynamically arrested phase separation.

Minor comments

1. Why use a binary Flory-Huggins implementation when this is so clearly a ternary system? The authors may wish to examine the work of Dan Deviri or Hue Sun Chan in terms of the ternary analytical ternary model, but it strikes me as odd to make this approximation, given the effort expended on including an electrostatic contribution to the free energy of mixing; is this really a more important feature of the system than correctly describing the molecular nature of the components? All this said I’m unconvinced this is actually a major issue and is not something I think is essential for revisions; I was just surprised!

We agree that the model system is ternary and we are aware of the extensive theoretical advances made by Dan Deviri, Hue Sun Chan, Glenn Fredrickson and Rohit Pappu on multicomponent systems. A central difficulty in employing the more complex theories is that the number of fitting parameters grows rapidly as more polymeric species are added, and prediction of phase boundaries becomes extremely computationally costly and sensitive to small changes in the long-chain regime. Translating the ternary phase boundary to a single component boundary further involves computation of tie-lines, and as the experiments performed here are over a few temperatures this whole process has to be repeated with temperature effects factored in - which, again, is not a trivial point at all. Explaining the formation of trapped liquids, however, does not require these theoretical complications. To illustrate

this point, we have added minimalistic simulations of the ϕ^4 theory and reproduced the micro-syneresis by simply changing the simulation temperature (Fig. 5a and supplementary text 2). We feel that simplicity contributes to explaining our findings and the mechanism, over introducing extensive physical details.

2. I would be cautious in interpreting the FRET efficiency data directly (Fig S1); given lifetime and quantum yields should also all be temperature-dependent and the change in FRET efficiencies is ~ 0.03 , at the very least a control (PEG in dilute conditions) and error bars would be critical to interpret these data.

We thank the reviewer for this comment. We are aware that the lifetime and quantum yields are also temperature dependent and have for this reason performed the control the reviewer described. We compare the lifetime in the presence of the acceptor, to the lifetime of the donor in absence of the acceptor at each of the temperature of interest (both τ_{DA} and τ_D are measured at the 20, 35 and 55 °C). Secondly, the reviewer states that error bars would be critical to interpret this data and we have thus added these.

3. I expected more in terms of interpreting the physical meaning from some of the parameters. Prior literature There was quite a bit of – in my opinion – absolutely central literature missing. 1. Boeynaems et al. 2019 PNAS explicitly discuss deep quenching into the two-phase regime for protein:RNA mixtures driving inhomogeneities as kinetically arrested systems, although I emphasize this work only helps support the current study. 2. Kaur et al. Nature Comms. 2021 directly examine many of the same questions; this is an important paper to cite and discuss. 3. Fisher et al. Nature Comms. Also report this phenomenon and explore a wide variety of parameters. It would be important for the authors to

We thank the reviewer for sharing this literature. We agree this is very relevant literature and have added this (line 36, refence 12-14).

We thank the reviewer for asking questions and sharing their feedback on this manuscript, which we believe they have taken great care in doing.

Reviewer #3 (Remarks to the Author):

In this manuscript, Erkamp and colleagues describe in vitro experiments with PolyA RNA + PEG showing that condensates adopt multiphase structures via a kinetically arrested phase transition. They show that cavities form at 20C and dissolve at 55C, in a reversible manner. By showing reduced cavity formation at lower cooling rates, they show that cavitation is a kinetic process rather than a thermodynamic one. In order to analyze the composition of fluid within the cavities, capillary velocity was evaluated by measuring the aspect ratio, width, and length of various cavity fusion events. Cavities were ruled to be similar in composition to the bulk dilute phase. Binodal and spinodal curves were calculated using multiple models, and the formation of cavities is represented by deviations from the binodal towards the spinodal, due to slow diffusion of PEG and PolyA. Finally, reconstituted stress granules show cavity fusion and enlargement over time, even without addition of RNA or temperature change, pointing to a more permanent gel-like behavior.

This work contributes to the effort of understanding biomolecular condensates, which is a very timely topic. The authors make strong arguments for the formation of cavities as an inherent,

kinetically driven process in their in vitro system, which is interesting. However, the biological relevance of this observation is unclear, with the size, composition, the condensate environment and activity being far different from in vivo conditions. Because of this concern, I think the paper is better suited for a specialized journal and I cannot recommend publication in Nature Communications.

We thank the reviewer for taking the time to give feedback on our manuscript, their observations and feedback have helped us to improve our manuscript significantly. We have performed additional experiments on biologically relevant condensates, reconstituted stress granules, and provide a response to all their helpful comments below.

Major Concerns:

1. As mentioned above, the biological relevance of the presented observations is unclear: The authors report that cavities only form in large condensates not in the small ones. For example, Fig 2 shows that at a condensate radius of 14 μm , cavities are never formed. While this size is small relative to the other condensates studied in this work, it is huge compared to the condensates found inside the cell. Given that most cellular biomolecular condensates are one micron or smaller in size, it suggests that such cavitation could not occur in condensates inside the cells.

Further, the observed diffusion rate of PEG is $\sim 12 \mu\text{m}^2/\text{s}$, much faster than the $\sim 1 \mu\text{m}^2/\text{s}$ measured for in vivo condensates, which could explain the lack of cavitation at small sizes. However, Extended Data Fig 9 shows a diffusion rate of $2.5 \mu\text{m}^2/\text{s}$ for PolyA, much closer to true biological regimes, which again raises question of biological relevance. A discussion is needed of why polyA condensates do not form cavities at less than a 14 μm radius. The effect of size scaling and the translation of the observations to small biologically relevant sizes should be discussed.

The reviewer makes an important observation, the enclosed droplets are only formed when condensates of sufficient size with slow diffusion undergo quick enough changes. Only in this case does the system deviate from the binodal, towards the spinodal. In our model system, we observe that condensates of 14 μm do not form enclosed liquids at any of the tested cooling rates. This seems to indeed reduce the biological relevance of the mechanism, since the most condensates inside of cells are much smaller than this. However, this can be counteracted by the slower diffusion rate that condensates in cells have. As the reviewer points out, the diffusion rate in condensates in cells is closer to $\sim 1 \mu\text{m}^2/\text{s}$. In figure 4B, we see that the diffusion rate effects the location of the boundary between forming enclosed dilute liquids and not forming enclosed dilute liquids significantly. The boundary shifts downwards, meaning that an enclosed dilute phase inside condensate form more easily.

Figure for reviewer: Figure 5c, but showing the expected boundary at 12 and 1 $\mu\text{m}^2/\text{s}$ (black line).

Notably, rather than temperature changes as reported on the x-axis, condensates in biological systems might encounter a variety of different events that change their composition, like concentration changes of compounds or pH changes. We have chosen our model system to be able to investigate how this internal structure can be formed and explain the observations done in cells and in vitro. Indeed, as the reviewer points out, a translation from the observations in our model system to biologically relevant systems is required. We have performed additional experiments on reconstituted stress granules (Fig. 4) and have added a model (Fig. 5a and supplementary video's 2, 3, 4 and 5) to make this translation easier.

2. In addition, the smaller condensates might be dominated by surface effects, which are negligible for large in vitro condensates. This should be considered, as this can cause different types of behavior at small sizes, which are biologically relevant.

We thank the reviewer for this comment, which made us carefully consider the importance of surface effects. We have calculated what percentage of molecules is at the interface of the condensate with the surrounding solution (y-axis) as a function of the radius of the condensate (x-axis) and on the biomolecules making up the condensate (colour).

Figure for reviewer: Varying the radius of the condensate and the biomolecules inside, we plot the number of molecules that are at the interface as a percentage of the total.

Based on this, we find indeed, like the reviewer said, that small condensates are dominated by surface effects. For larger ones, the amount of molecules at the surface very quickly drops off. Notably, this depends on the radius of the biomolecules as well. For example, PEG20kDa, which was used in our study, has a hydrodynamic radius of 3.01 nm (table 2 of [1]). Even if the average hydrodynamic radius of biomolecules in the condensate was 10 nm (blue curve), the amount of molecules at the surface would still be below 20 % for a condensate with a diameter of 0.1 nm. Since we are interested in condensates large enough to contain a trapped droplet of liquid, it seems unlikely that surface effects would alter our proposed mechanism.

[1] Dong, X; Al-Jumaily, A.; Escobar, I. Investigation of the use of a bio-derived solvent for non-solvent-induced phase separation (NIPS) fabrication of polysulfone membranes. *Membranes* 2018, 8(2), 23

3. The authors observe that the cavities have similar/same composition as the surrounding solution of the condensates. This is in contrast to in vivo condensates, whose subcompartments are biopolymer-poor, but contain yet another biomolecule that makes up the cavity, in other words they are of different composition than the surrounding solution. If indeed these cavities form initially simply with the dilute phase, then the necessary biomolecules would have to pass through the dense phase into the newly formed cavities. This would disrupt the kinetics of the condensate and possibly terminate the kinetically arrested phase transition. So, it is not immediately clear, how to apply the observations of this paper to in vivo systems.

Indeed, in cells, we have a different situation from our model system. In our model system, condensates are rich and the dilute phase is poor in PolyA and PEG. When the condensate changes composition, droplets of dilute phase can be placed on the inside of the condensate, creating a liquid from the compounds already present in the condensate.

Now we consider the situation in a cell. The “dilute phase” of a cell, for example the cytosol, is described as a dilute phase in contrast to the condensate, but notably contains a significant amount of different biomolecules. Another important difference between a cell and our model system is that the cells is performing active processes. The cytosol is not a homogeneous, well-mixed solution. Matching with the reviewers comment, when we create a “dilute phase” inside of the condensate from the material already present in the condensate, we will create a liquid that will contain slightly different compounds than present in the surrounding “dilute phase”. After applying the mechanism described in the paper, we see that the observations in cells match with the proposed mechanism.

4. The authors address the in vivo relevance of Extended Data Figure 10, but the data presented is quite minimal. Although the Materials and methods section describes in detail the complex procedures for preparation of the reconstituted stress granules and antibody-DNA condensates, there is no analysis presented, only a presence of cavities is shown. The kinetics of these cavities is not analyzed in a quantitative way, which would support the claims in this work. Additionally, the size regimes are still an order of magnitude above those in in vivo biomolecular condensates. In fact in Extended Data Figure 10a, it is clear that the smallest droplets do not form cavities. Since these are the most biologically relevant cavities, some discussion should be focused on them and corresponding kinetics analysis of this data should be added.

We thank the reviewer for their comment. Indeed, the main focus of our manuscript is the discovery of the general mechanism for forming condensate double emulsions. To provide more in dept information about the reconstituted stress granule system and quantify this, we have added Figure 4 and Ext. Data fig. 7 to the manuscript. Here, reconstituted stress granules are prepared (left side of figure). Then, we have added the PolyA RNA to the solution, the condensate changes composition as it takes up some of this new material and thus forms trapped dilute droplets, poor in both G3BP1 protein and nucleic acids, shown by the newly added dye (figure 4a and Ext. data fig 7). We can now see that a composition change results in the formation of this internal structure. We have quantified the number of trapped droplets before and after the addition of PolyA and dye as a function of the radius of the condensate in figure 4b. This shows that the number of trapped droplets is highly dependent on the size of the condensate, as was also observed in the PolyA-PEG system (lines 147 – 155).

5. Another difference between the presented in vitro condensates and in vivo condensates is the presence of activity in the latter. This is only very briefly mentioned in the manuscript, but is a central point of living systems and hence should be adequately addressed. Could authors elaborate how would activity impact their presented picture?

We thank the reviewer for their comment on what the impact of activity would be. We have investigated whether activity could influence the phase diagram (next paragraph) or if it could

influence the ability of the system to move away from the binodal and nucleate dilute phase (paragraph after that).

In literature, both motility-induced phase separation [1, 2] and phase separation of chemically active species [3, 4] are discussed. The general theoretic approach follows a similar framework as the static picture. One derives an 'effective' free energy density that only depends on polymer concentrations to absorb the interaction/reactions into coefficients in the free energy expression, for example see equation 27 in [1] and equation 6 in [4]. These more detailed models represent higher order correction terms on top of the mean-field Flory-Huggins picture. Notably, they do not change the general behaviour of the system qualitatively. Specifically, the spinodal concentration (corresponding to local instability) and binodal concentration (corresponding to global instability) stay the same and are calculated from the 'effective' free energy density.

Dynamically arrested phase separation as discussed in this manuscript relies on the crossing of the dense phase into the spinodal region, to nucleate the dilute phase in the dense phase. While we find that the location of the binodal and spinodal is not affected by activity, the activity can still influence if trapped liquids are formed by influencing the environment. Specifically, activity, like a sudden increase or decrease in the concentration of a compound can cause composition changes in the condensate, which in term can cause kinetically arrested phase separation. We observe this for example in the stress granule system, where RNA addition leads to the formation of trapped droplets. We appreciate this comment from the reviewer, as it made us consider that the activity in cells might promote dynamically arrested phase separation.

- [1] Cates, M. E., & Tailleur, J. (2015). Motility-induced phase separation. *Annual Review of Condensed Matter Physics*, 6(1), 219–244.
- [2] Wittkowski, R., Tiribocchi, A., Stenhammar, J., Allen, R. J., Marenduzzo, D., & Cates, M. E. (2014). Scalar ϕ^4 field theory for active-particle phase separation. *Nature Communications*, 5, 1–9.
- [3] Longo, T. J., & Anisimov, M. A. (2021). *A Mean Field Theory of Phase Transitions Affected by Molecular Interconversion*. Arxiv.
- [4] Li, Y. I., & Cates, M. E. (2020). Non-equilibrium phase separation with reactions: A canonical model and its behaviour. *ArXiv*.

6. In Extended Data Fig 2, both PolyA and PEG signals are shown. In Fig 1-3, there is no information provided if the signal shown is polyA or PEG. Based on the shape of cavities shown in Extended Data Fig 2, it would appear that the cavities are more spherical in PEG, which is very similar to the signal shown in Fig 1-3 and thus suggest we are looking at PEG signal. Yet, the phase diagram is based on the PolyA concentration. This should be clarified. It would be helpful to see some more examples of PolyA cavities. They seem less spherical than the corresponding PEG cavities, is there a critical size where polyA doesn't form cavities but PEG does? The work would be improved by a discussion on the differences in cavity structure in the two signals. Figures should also be clearly labeled to indicate whether it is PolyA or PEG being imaged.

We thank the reviewer for this comment. Indeed, the description of figure 1-3 should state if the signal shown corresponds to PolyA or PEG. We have made changes in line 77, 84 and 106. PEG is only shown in Extended Data Fig. 2, to highlight that they are both PolyA and PEG are enriched in the dense phase. This is highlighted in line 376.

7. In addition to different appearance of cavities in polyA vs PEG, their diffusion rates are also quite different ($2.5 \text{ } \mu\text{m}^2/\text{s}$ vs $12 \text{ } \mu\text{m}^2/\text{s}$). The authors use pictures of PEG and polyA interchangeably, implying that wherever there is a PEG cavity there is also polyA cavity. However with such different diffusion rates, the kinetics of PEG and polyA may affect the cavity formation. To address this, both signals in Fig 1-3 and in Extended Data Fig 9 should be shown.

We thank the reviewer for their comment. Indeed, trapped droplets are depleted in both PolyA and PEG. Besides Ext. Data Fig. 2, all pictures in the manuscript of this model system show PolyA, which we have clarified (see comment above). We have chosen to highlight that these compounds are depleted / enriched in the same location, rather than showing both the PolyA and PEG image for every figure, since they are very similar looking images.

Additionally, the reviewer comments on the difference in diffusion rate of PEG and PolyA, which we believe to be because of the difference in the size/hydrodynamic radius of PEG and PolyA. Notably, the PolyA used in this study is quite polydisperse (2100 - 10600 subunits or 700 - 3500 kDa), making the diffusion coefficient of PolyA, more difficult to interpret than that of PEG. For this reason, we have removed the graph of the FRAP of PolyA and have focussed on the diffusion of PEG.

Lastly, the reviewer points out that trapped droplet formation takes place while compounds with different diffusion rates are in the condensate. We agree with this and think this is very relevant for condensates in cells, which, like the reconstituted stress granules, can be made up of a wide range of compounds with different diffusion coefficients.

8. Fig 3 discusses the fusion rate of cavities within condensates, but the size of these cavities is not discussed. Do cavities fuse at constant rates in all scenarios, or are they dependent on condensate size or individual cavity size?

We thank the reviewer for this comment. We have observed that the size of condensates or cavities influences the fusion rate significantly. In the original figure 3 b, c and d (now figure 3 d, e and f), we study the fusing of condensates, trapped droplets and trapped droplets with the surrounding dilute phase respectively. The average diameter of the liquids fusing is shown on the x-axis. We have changed these labels to d_{avg} or d (μm) to make this more clear. This figure also answer the question of the reviewer: Yes, we find a positive correlation between the diameter and fusion time. The slope of this fitted line is the inverse capillary velocity, which we compare.

Notably, the reviewer refers to the events as cavities “fusing” rather than “merging”. We agree with this description and have thus changed terms like “merging” “merge” etc. to “fusing” “fuse” etc. throughout the manuscript.

Minor Issues:

1. In Figure 2, the temperatures in the subscript of radius in the top row are too small to be easily legible.

2. Lines 57, 340: form is misspelled as “from”

We have removed the temperature from figure 2 and added this to the figure description (line 81).

We have changed this spelling of “from” in line 69. The other typo was found in a line removed from the manuscript.

We thank the reviewer again for taking the time to give such fantastic feedback on our manuscript.

Reviewer #4 (Remarks to the Author):

In this work, the authors present results from measurements of synthetic condensates and demonstrate that vacuole (or cavity) formation is a dynamically controlled phenomenon. The results are presented for a ternary mixture comprising solvent, PEG, and poly-rA. The results are very interesting and likely to be of broad interest and relevance.

We thank the reviewer for taking the time to give feedback on our manuscript. We appreciate their observations and comments, which have helped us to improve the manuscript.

While a series of interesting findings are presented, what is lacking is two-fold. The phrasing (see specific comments below) of various aspects of the study and the lack of distinction between spatially organized multiphase structures in multicomponent systems vs. vacuole formation (observed in yeast nucleoli) is not clearly made. This is relevant because the extant literature clearly lays out the distinction between multiphase structures coming about from thermodynamic control vs. vacuolized structures forming via dynamical control. Second, a very interesting body of data is presented. However, a robust theoretical or phenomenological computational model is what is needed to tie things up. This is missing. If these changes can be made, then this would make for a very interesting and timely contribution that deserves to be published in Nature Communications. The specific comments are categorized as major and minor.

We thank the reviewer for highlighting their 2 most important points of feedback. We appreciate the importance of distinguishing between multiphase systems and double-emulsion condensates the manuscript focusses on. Indeed, the double emulsion structure is a result of dynamical arrest, while thermodynamic factors are very important for multiphase structures. Thus, it is important to introduce both concepts and then point out that these are different phenomena.

Secondly, the reviewer points out that a robust theoretical computational model would tie this work up well. Thus, we have made this model and attached the results to this report. Below, we offer a point-to-point response to all the comments and these 2 points in more detail.

Specific major comments

1) Distinction between multiphase, spatially organized structures vs. structures with holes i.e., vacuoles. The introductory narrative takes a bit of parsing because it is not clear if the authors are referring to multilayered structures such as nucleoli or if the focus is on observations (less in biology and more in the synthetic literature) of vacuoles that appear to create the impression of coexistence of filled and hollow phases. Please consider rewriting the introduction to clarify the problem of interest.

Once this is done, and if one does introduce the topic of true core-shell architectures, then please note that there is a well honed thermodynamic basis for the appearance of such structures. This was first put forth in the work of Feric et al., ([http://www.cell.com/cell/fulltext/S0092-8674\(16\)30492-5](http://www.cell.com/cell/fulltext/S0092-8674(16)30492-5)). Please note that even the simplest ternary mixtures of polymers can have core-shell and vacuolar structures. These have been explained on the basis of sequence-specific immiscibility. In addition to the work of Feric et al., and Fei et al., there is the work of Harmon et al., which shows just how one can get spatially organized droplets out of differential solvation effects. Please see: <https://iopscience.iop.org/article/10.1088/1367-2630/aab8d9>. Likewise, Simon et al.,

show this using ELPs of different hydrophobicities <https://www.nature.com/articles/nchem.2715>. The key message is that there is a solid thermodynamic framework for spatially organized droplets. However, the presence of vacuoles and the emergence of vesicles is less clear in terms of the interplay between thermodynamic vs. dynamical control. The work of Boeynaems et al., (which showed a thermodynamic basis for the spontaneous emergence of spatially organized structures) does demonstrate how dynamical arrest can interfere with phase separation, as first illustrated by Sciortino et al. Please see: <https://www.pnas.org/content/early/2019/03/28/1821038116> and references therein.

Further, in a recent study, Seim et al., have demonstrated that "off-pathway" oligomerization via homotypic interactions can impact the protein density and even the rearrangement dynamics of protein-RNA condensates that form mainly via heterotypic interactions. Please see: <https://www.pnas.org/doi/10.1073/pnas.2120799119>. The key thrust of the point being made is that spatially organized condensates can arise, and possibly do arise, from purely thermodynamic considerations. Vacuole formation continues to be a puzzle, and it is satisfying to see a more intuitive dynamical argument being put forth here. These points, with appropriate nods to the literature, could be better clarified in the narrative, both in the introduction and in the connections made between the observations and the literature. As currently crafted, some of the verbiage is stilted and this is likely to engender a lot of confusion. In this context, it is perhaps a stretch to assert that "While this class of structure is thus commonly encountered in nature...". It is not clear that such structures are common in nature.

We thank the reviewer for this comment considering how to place this manuscript in context with previous work and for sharing many relevant paper with us. The reviewer requests that we carefully consider that there is an established thermodynamic framework for obtaining multiphase condensates. In fact, purely thermodynamic factors can cause the formation of many of the different condensate structures that have been found (for an overview of these, we like figure 2 of Fare et al. <https://royalsocietypublishing.org/doi/10.1098/rsob.210137>). Sequence-specific immiscibility, solvation effects and differences in hydrophilicities can be important for this.

Indeed, much less is known however about the role of kinetics in the structure of condensates and about the presence and formation of a phase poor in biopolymers inside the condensates. Indeed, our manuscript puts forward a mechanism combining these.

We have added many of the papers shared by the reviewer to the manuscript. Feric et al. (ref 10), Harmon et al. (ref 11), Boeynaems et al. (ref 12), when multiphase condensates are discussed. Simon et al (ref 25) is referenced where we have now highlight how kinetics underly the structure in this work, rather than thermodynamics. We have also changed the part "While this class of structure is thus commonly encountered in nature..." (line 42).

2) At the risk of seeming like a broken record, please clarify that the structure being investigated is that of a condensate with holes i.e., a condensate undergoing vacuolization. This is super relevant because as the authors clearly note, they are studying the formation of "cavities, liquids poor in PolyA and PEG inside the condensate". Indeed, even the title is misleading. It should be something like: Condensates with holes form via dynamical arrest or something like this because almost every reader will jump to the conclusion that facsimiles of nucleoli arise due to dynamical arrest, and this is clearly not the case.

We absolutely agree with reviewer and thank them for bringing this up. We think changing the term “multiphase”, often used to refer to the structure of condensates like nucleoli can help to clarify what structure we are looking at significantly. Instead, we think the term “double-emulsion” condensates is more suitable. “multiphase” also implies that more than 2 phases are present, while we notably have just 2, with the dilute phase trapped in the dense phase. “double-emulsion” does not imply that more than 2 phases are present.

Additionally, to clarify the structure we are investigating, we have also changed the term “cavity” which might remind some readers of air pockets, rather than a space with a liquid. We instead refer to “trapped droplet(s)”.

3) The relevance of the work of Simon et al. (see above) cannot be overemphasized.

We agree and have added this reference to the manuscript (ref 25).

4) This particular statement "If cavity formation is a kinetic process we would expect cavities to form at fast composition changes, whereas cavities would be formed at slow composition changes if it is a thermodynamic process" is concerning because the standard definition of thermodynamic control is of reversibility and reproducibility no matter the starting point. It appears the statement is intended to distinguish between quenched disorder vs. the lack of such disorder upon annealing. This interpretation is consistent with the data in Figure 2, which are really neat. Please consider rewording.

We agree and have changed lines 68-70 accordingly.

5) To give credit where it is due, Banerjee et al., did make the point that prior to dissolution, the peptide-RNA condensates they studied go through vacuole formation. Essentially they viewed this as nucleation of the dilute phase within the condensate. I refer here to reference 12, which could be better cited and integrated in with the current results.

We have changed line 38-39 to match with this.

6) Please elaborate on the details of the "three classes of fusion events".

Indeed, the three different fusion events, (1) fusion of condensates, (2) fusion of trapped dilute droplets and (3) fusion of trapped droplets with the dilute phase surrounding the condensate, are shown in the figure 3d, e and f schematically, but were not mentioned as such in the text. We have added this in lines 122-123.

7) This sentence is difficult to parse because the meaning of it is not clear: "Since the viscosity entering the capillary velocity equation is always that of the dense condensate phase, any differences show differences in the surface tensions between the different interfaces." Please reword and clarify.

We thank the reviewer for bringing this up. We rewritten this section to more clearly explain how we come to the conclusion that the surface tensions are similar (line 122-144). The sentence in question has been removed.

8) The current system is a ternary mixture comprising the solvent, PEG and poly-rA. If the

concentrations of the macromolecules are variables, and so is the temperature, then the phase boundary is not longer a binodal. This verbiage is problematic because it is misleading. It appears that the authors are describing a mechanism where the solvent activity is fixed as is the concentration of PEG. Then the variables are the concentration of poly-rA and temperature. Is this the case? It appears to be the case based on Figure 4A. If so, this should be specified very clearly. However, it is then incumbent to specify that for fixed amounts of PEG, the concentration titrations of poly-rA cause significant sub- and super-stoichiometric regions to be explored along the phase diagram. Inasmuch as this is the case, and given that the ratios of molecules that combine to drive phase separation are viewed as the relevant order parameters for vacuolization and / or reentrant phase behavior, the stoichiometric ratios and their connection to cavity / vacuole number and size become super important and relevant.

We agree with the reviewer that it is important to be clear what variables are chosen in each experiment. For completeness: In figure 1, only temperature is varied (rows). In figure 2, condensate size (columns) and cooling rate (rows) are varied. Pictures of the beginning (approximately halfway through) and at the end of cooling are shown. In figure 3, the time changes (row). In figure 4A, indeed as the reviewer says, the variables are temperature and Poly-rA concentration. We have clarified this by changing figure 4A to the following:

In this phase diagram, instead of “interaction strength” we have specified that we are changing the temperature. Additionally, because both the dilute phase and dense phase “branch” of the diagram are shown, it becomes easier for the reader familiar with these kinds of phase diagrams to orientate themselves in the figure. On the x-axis, we have the total Poly-rA concentration. Indeed, in this figure, the PEG concentration is constant (5 w/w %) and we have specified this in line 361. We agree with the reviewer that the stoichiometric ratios of Poly-rA and PEG are very relevant for the phase behaviour and kinetically arrested phase separation.

9) Overall, the lack of a theoretical framework or a computational model that explains the data leave one wanting more. Given that cavity formation can be observed in silico, one can then investigate the effects of a deep or shallow quench and timescales in dynamics simulations or move sets in MC simulations. Examples of this abound in the soft matter and biological phase separation literature. Right now, the MS ends with a set of observations, a somewhat arbitrary classification of viscoelastic vs. liquid-like materials (should this not be dependent on the dynamical moduli), and a proposal that reads a bit like an assertion. Having a model that incorporates the measured parameters or a simulation that captures the phenomenology would be considerably more satisfying.

We agree with the reviewer that a model to show kinetically arrested phase separation would be very interesting and could potentially improve our understanding further. We have constructed such a model. We have modelled the following situations:

Supplementary video 2 - Condensate formation. Initially, a homogeneous mixture is shown, before phase separation takes place, forming a dilute and dense phase. Notably, the dense liquids merge with each other and form a circular liquid over time. A cross-section is shown as well.

Supplementary video 3 – Lowering temperature quickly. By reducing the temperature, the ideal density in the dense phase increases. Some of the condensates reach this density by creating a dilute phase inside of the condensate. Notably, depending on the size of the condensate, either 0, 1 or multiple trapped liquids are formed. This matches well with our observations in figure 2. This also matches our observations from figure 4B, where we see that there is a critical size for the formation of trapped dilute phase. Additionally, we observe that the trapped liquids can fuse, similar to what was observed in figure 3. Interestingly, we see that additional small condensates are created during this stage.

Supplementary video 4 – Lowering temperature slowly. While the composition of the dilute and dense phase changes similarly to that in video 3, no trapped droplets are formed inside of the condensates, because the change occurred slower.

Supplementary video 5 – Increasing the temperature. Here, we increase the temperature, effectively reversing the composition change back to the original state and in the process remove trapped dilute liquids.

The picture above was added to Figure 5 the video's are added as the supplementary materials and supplementary text 2 explains the model in more detail, as is done below. We really appreciate this comment from the reviewer, as adding this model highlight that the mechanism discussed in the paper applies generally to phase separating systems, undergoing a composition change.

Details about the model:

We have created this model based on reports on micro-syneresis [1-3]. This simple model uses 2-dimensional ϕ^4 theory [4-8] and observed cavity formation with a suitable choice of parameters. Denote the two spatial coordinates as x and y , we define the field $\phi(x, y)$ and the total free energy functional $\mathcal{F}[\phi, \nabla\phi]$ using the expression

$$\mathcal{F}[\phi, \nabla\phi] \equiv \iint dx dy \left[a\phi^2 + b\phi^4 + \frac{k}{2} |\nabla\phi|^2 \right]$$

, with the first two integrands denoting the bulk energy and the third term surface tension [9]. The parameters (a, b, k) are set in simulation. The current density J is then defined as

$$J \equiv -M\nabla \frac{\delta\mathcal{F}}{\delta\phi} + \sqrt{2DM}\Lambda$$

Where M is a mobility constant, D the thermal noise temperature and Λ a unit-variance Gaussian noise. The functional derivative $\frac{\delta\mathcal{F}}{\delta\phi}$ is the chemical potential and can be calculated as

$$\frac{\delta\mathcal{F}}{\delta\phi} = 2a\phi + 4b\phi^3 + k\nabla^2\phi$$

In the bulk picture without surface tension, phase separation occurs when $a < 0$ and $b > 0$ with binodal concentrations $\phi_{\text{bin}} = \pm\sqrt{-\frac{a}{2b}}$. Lowering the system temperature is equivalent to lowering the value of a as a is often assumed to be proportional to $(T - T_c)$, with T_c the Curie temperature below which phase separation occurs. To compute the time evolution of the field ϕ we use the continuity equation

$$\dot{\phi} = -\nabla \cdot J = M\nabla^2 \frac{\delta\mathcal{F}}{\delta\phi} - \sqrt{2DM}\nabla \cdot \Lambda$$

The simulation uses the established code [7] and we use a 2D simulation box with 128 by 128 cells, each of unit side length. The time step is set to $dt = 0.001$. The field $\phi(x, y)$ is initialised with a mean of -0.1 and a random uniform noise between 0.05 and -0.05 . The simulation is separated into three stages matching with the video's provided (1) Condensate formation (Incubation), (2) Decreasing temperature and (3) Increasing temperature. In all stages we keep the parameter values

$$\begin{aligned} D &= 0.2 \\ M &= 1 \\ b &= 7 \end{aligned}$$

And for individual steps the parameters are, with t denoting the instantaneous simulation time point and T the total simulation time:

(1) Condensate formation:

$$\begin{aligned} a &= -1 \\ k &= 3 \\ T &= 2000 \end{aligned}$$

(2) Decreasing Temperature:

$$\begin{aligned} a &= -1 - 19 * \frac{t}{T} \\ k &= 3 + 37 * \frac{t}{T} \\ T &= 50 \end{aligned}$$

(3) Increasing Temperature:

$$\begin{aligned} a &= -1 \\ k &= 3 \\ T &= 10 \end{aligned}$$

In short, after the initial incubation, the quadratic term $\alpha\phi^2$ is rapidly decreased to drive the system into an unstable regime. The surface tension is increased accordingly to keep up with the bulk energy scale. In the “Increasing temperature” phase the system moves towards a condition similar as at the end of the “condensate formation” stage.

- [1] Tanaka, T., Hocker, L. O., & Benedek, G. B. (1973). Spectrum of light scattered from a viscoelastic gel. *The Journal of Chemical Physics*, 59(9), 5160–5183.
- [2] Tanaka, T., Sato, E., Hirokawa, Y., Hirotsu, S., & Peetermans, J. (1985). Critical kinetics of volume phase transition of gels. *Physical Review Letters*, 55(22), 2455–2458.
- [3] De Gennes, P.-G., & Gennes, P.-G. (1979). *Scaling concepts in polymer physics*. Cornell university press.
- [4] Singh, R., & Cates, M. E. (2019). Hydrodynamically Interrupted Droplet Growth in Scalar Active Matter. *Physical Review Letters*, 123(14), 148005.
- [5] Bray, A. J. (1994). Theory of phase-ordering kinetics. *Advances in Physics*, 43(3), 357–459.
- [6] Cahn, J. W., & Hilliard, J. E. (1958). Free Energy of a Nonuniform System. I. Interfacial Free Energy. *The Journal of Chemical Physics*, 28(2), 258–267.
- [7] Singh, R., & Cates, M. E. (2019). Hydrodynamically Interrupted Droplet Growth in Scalar Active Matter. *Physical Review Letters*, 123(14), 148005.
- [8] Bray, A. J. (1994). Theory of phase-ordering kinetics. *Advances in Physics*, 43(3), 357–459.
- [9] Cahn, J. W., & Hilliard, J. E. (1958). Free Energy of a Nonuniform System. I. Interfacial Free Energy. *The Journal of Chemical Physics*, 28(2), 258–267.

Specific minor comments

- 1) Please note that paraspeckles are not condensates in the conventional sense. They are micelles that undergo sphere to rod transitions. Please see:
<https://www.embopress.org/doi/full/10.15252/emboj.2020107270>.

We thank the reviewer for sharing this work. We removed this example from line 35.

- 2) Given the focus on spatial inhomogeneities within condensates, the prefix of LL in LLPS is misleading. The term serves as a straw-man, which almost all reasonable enthusiasts and data will take down. Please consider deleting all mention of LL and / or liquid-liquid, at least in the context of biological phase separation. For the synthetic system here, it probably makes sense. Also, when discussing liquids, please specify what types of liquids one should conjure up as a models.

We have changed “LLPS” to “PS” in line 15.

- 3) As a matter of taste "binodal curve" should be just binodal.

We agree with this and have changed “binodal curve” to “binodal” throughout the manuscript (for example line 188).

We thank the reviewer again for taking the time to give feedback on our manuscript. We particularly appreciate the literature and suggestion to add a model.

REVIEWER COMMENTS

Reviewer #1 (Remarks to the Author):

This revised study by Erkamp, Sneideris et al. is much improved. They address many of the salient questions I and other reviewers raised through a combination of new analyses/experiments, better presentation, and reprioritization of existing data that removed poorly supported statements.

I now broadly support publication of this exciting study. I recommend a few minor changes that I believe can improve paper presentation but do not require them i.e. leave them at the author's discretion:

1. use of consistent terminology - the authors flexibly use "trapped droplets"/ "double-emulsions"/ "vacuoles".
2. In the section on "scaling laws" (Fig 3b-c) - the text is unclear it says the data fits "expectations". Spelling out the model to specify what the expectations are (for e.g. from LSW) is needed.
3. A general comment - the authors have done a great job of trying to be succinct but in many subsections it feels like they may have overdone this. Given the lack of word limits, I would suggest expanding text to add details on specifying more about simulation model, more about scaling laws, and explaining some of the interesting results in Ext. Data, particularly 6,7, and 10.

Reviewer #2 (Remarks to the Author):

The authors have done an outstanding job in addressing the reviewers' comments in significant depth. I am satisfied that my comments have been well addressed and congratulate the authors on an excellent study.

Reviewer #3 (Remarks to the Author):

I found the authors' response very confusing: While they responded to my comments in their response document, most of my concerns are NOT addressed in the revised manuscript in any way. For some of their responses, authors even prepared figures for only my viewing, yet no action was taken on the manuscript or SI. Hence, effectively, in the revised manuscript most of my raised issues remain completely unaddressed and thus the paper still suffers from the same shortcomings that I listed in my first review. I.e., the biological relevance of the presented observations is still unclear, with the size, composition, the condensate environment and activity being far different from in vivo conditions. Therefore, I think the paper is better suited for a specialized journal and I cannot recommend publication in Nature Communications.

Below I provide a few examples of points I rose in my original review and which were not addressed in the revised manuscript:

1. In my comment 1, the authors agree that the biological relevance is reduced due to the size minimum needed for cavity formation, and they speculate that the slower diffusion rates of condensates in cells could mitigate this. They provided an alternative plot for us showing the expected boundary at a slower diffusion rate. While this does address the raised point, it does not provide suitable evidence for biological relevance, simply a speculation. More importantly, this graph and any discussion on the lack of biologically relevant sizes is not mentioned anywhere in the manuscript. I believe that the potential in vivo applications of this work is by far the most important aspect of it, and any discussion centered around it should be included in the manuscript.

In addition, in the response to my comment 1 they do not address the fact that polyA, with a lower diffusion rate, seemingly has the same behavior as PEG. They do address this point in comment 7, stating that polyA is polydispersed and thus its diffusion coefficient is "more difficult to interpret." Given that all shown experiments show condensates containing both PEG and PolyA, it is important to see condensates formed from just one compound and comparing the size at which trapped droplets appear.

2. In response to my comment 2, the authors have provided a plot showing the percentage of molecules at the surface given a condensate and particle radius. While this plot makes it clear that their condensates are large enough to ignore surface effects, when considering biologically relevant scales - such as the 10nm particle size and 0.1 micron condensate size example provided by the authors - the percent of molecules on the surface is between 10-20%. This is a significant portion and further calls into question the in vivo relevance of this work, given that at biologically relevant length scales the surface effects would be non-negligible. This is a critical point that needs to be addressed in the manuscript but is not. Again, neither the graph that authors show, nor any discussion of this point is mentioned anywhere in the manuscript.

3. It is unclear what the authors meant in their response to my comment 3. To better recreate the environment of the cell, they describe creating a liquid with slightly different compounds present in the surrounding dilute phase. It is unclear if these are future experiments or ones that have already been done. If it is the latter, then those results should be described further. Clarification is needed. In any case, the authors have addressed my point here through their response to another reviewer, where they changed their terminology from "multiphase" to "double emulsion". This suitably distinguishes their condensates from multiphase biological structures such as nucleoli.

4. The authors have added a new figure illustrating and quantifying the effect of polyA addition to stress granules. While this is an interesting experiment using biological analogs, there is still a minimum radius needed to create the double emulsions, around 5 microns. This is still far larger than is biologically relevant, and no discussion is presented in the manuscript that addresses this point for potential readers.

5. In my comment 5, I raise question about activity, which also was not addressed anywhere in the manuscript.

6. In my comment 6, the authors made appropriate changes in the labelling of PolyA and PEG, but did not address if there is a critical size where polyA does not form cavities but PEG does.

7. For comment 7, the authors stated that the PolyA used was polydisperse (700-3500kDa), making analysis of diffusion coefficients more difficult. While this does provide an explanation as to why the more biologically relevant molecule was not used, it does not provide any explanation as to why other slow diffusing compounds were not attempted. Given that using a compound with a diffusion rate closer to $1 \text{ } \mu\text{m}^2/\text{s}$ would dramatically increase the biological relevance of this paper, an explanation should be included into the manuscript.

The authors also included simulations of rapid or slow composition changes. I think that these simulations could be used to extend this work's biological relevance, by simulating smaller condensates made up of compounds with slower diffusion rates. If double emulsion formation cannot be achieved under those simulation conditions, that should be addressed as it impacts the application of this model to smaller biological systems.

Reviewer #4 (Remarks to the Author):

The authors have responded to all the comments and concerns I raised. Their responses and revisions exceed all reasonable expectations. This is important and timely work and should be published without further revisions. Nature Communications is the right forum for this work.

Reviewer #1:

This revised study by Erkamp, Sneideris et al. is much improved. They address many of the salient questions I and other reviewers raised through a combination of new analyses/experiments, better presentation, and reprioritization of existing data that removed poorly supported statements.

I now broadly support publication of this exciting study. I recommend a few minor changes that I believe can improve paper presentation but do not require them i.e. leave them at the author's discretion:

We thank the reviewer for their fantastic feedback and additional comments.

1. use of consistent terminology - the authors flexibly use "trapped droplets"/ "double-emulsions"/ "vacuoles".

We've removed vague terms like "multiphase" and now use "trapped droplets" to specifically discuss the droplets inside of the condensate and "double-emulsion" to discuss the structure. "vacuole" is only mentioned now because a cited article described their system as such (line 31, ref 15).

2. In the section on "scaling laws" (Fig 3b-c) - the text is unclear it says the data fits "expectations". Spelling out the model to specify what the expectations are (for e.g. from LSW) is needed.

Indeed, it is relevant to mention how the data is fitted in the main text, not just in supplementary information (added to lines 103-104, ref 28).

3. A general comment - the authors have done a great job of trying to be succinct but in many subsections it feels like they may have overdone this. Given the lack of word limits, I would suggest expanding text to add details on specifying more about simulation model, more about scaling laws, and explaining some of the interesting results in Ext. Data, particularly 6,7, and 10.

We have expanded Ext. Data 10 and the discussion on this figure in the main text.

Reviewer #2:

The authors have done an outstanding job in addressing the reviewers' comments in significant depth. I am satisfied that my comments have been well addressed and congratulate the authors on an excellent study.

We thank the reviewer again for taking the time to give such fantastic feedback on our manuscript.

Reviewer #3:

I found the authors' response very confusing: While they responded to my comments in their response document, most of my concerns are NOT addressed in the revised manuscript in any way. For some of their responses, authors even prepared figures for only my viewing, yet no action was taken on the manuscript or SI. Hence, effectively, in the revised manuscript most of my raised issues remain completely unaddressed and thus the paper still suffers from the same shortcomings that I listed in my first review. I.e., the biological relevance of the presented observations is still unclear, with the size, composition, the condensate environment and activity being far different from in vivo conditions. Therefore, I think the paper is better suited for a specialized journal and I cannot recommend publication in Nature Communications.

Below I provide a few examples of points I rose in my original review and which were not addressed in the revised manuscript:

We appreciate the opportunity to incorporate the reviewer's feedback in the manuscript further, which was indeed lacking in our last version. Specifically, we have added a more extensive version a graph previously shared just with the reviewer to the manuscript. Additionally, we now more

extensively discuss under what conditions the observations in the manuscript may have biological relevance (Ext. Data Fig. 10, lines 126-127, 176-180, 384-387, Ext. text 2, line 103-105). Below, we reply to the examples provided by the reviewer in more detail.

1. In my comment 1, the authors agree that the biological relevance is reduced due to the size minimum needed for cavity formation, and they speculate that the slower diffusion rates of condensates in cells could mitigate this. They provided an alternative plot for us showing the expected boundary at a slower diffusion rate. While this does address the raised point, it does not provide suitable evidence for biological relevance, simply a speculation. More importantly, this graph and any discussion on the lack of biologically relevant sizes is not mentioned anywhere in the manuscript. I believe that the potential *in vivo* applications of this work is by far the most important aspect of it, and any discussion centered around it should be included in the manuscript.

In addition, in the response to my comment 1 they do not address the fact that polyA, with a lower diffusion rate, seemingly has the same behavior as PEG. They do address this point in comment 7, stating that polyA is polydispersed and thus its diffusion coefficient is "more difficult to interpret." Given that all shown experiments show condensates containing both PEG and PolyA, it is important to see condensates formed from just one compound and comparing the size at which trapped droplets appear.

We appreciate this comment on the biological relevance and size of the condensates. To centre this discussion around *in vivo* observations, we have looked again at reference 20-24, papers in which the double-emulsion structure has been found in condensates in cells. When we examine their work, we see that this structure is observed in condensates with a diameter of for example 3 μm , 1 μm or 300 nm:

Schmidt, H. B. & Rohatgi, R. *In Vivo* Formation of Vacuolated Multi-phase Compartments Lacking Membranes. *Cell Rep.* **16**, (2016).

Kistler, K. E. *et al.* Phase transitioned nuclear oskar promotes cell division of *Drosophila* primordial germ cells. *Elife* **7**, (2018).

Arkov, A. L., Wang, J. Y. S., Ramos, A. & Lehmann, R. The role of Tudor domains in germline development and polar granule architecture. *Development* **133**, (2006).

Thus, this structure is indeed observed in condensates much smaller than the PolyA-PEG condensates that the manuscript mostly focusses on. Indeed, we believe this is due to a difference between the viscosity of the condensates we examined and those in cells.

Biologically relevant condensates often have a viscosity which is orders of magnitude larger than that of condensates containing mostly PEG. In the figure below, we can see this by comparing the green with magenta circle (figure from ref 34). Additionally, we observe that in vivo prepared condensates are often more viscous than those prepared in vitro. For example, NPM1 condensates are 2 orders of magnitude more viscous in vivo.

Wang, H., Kelley, F. M., Milovanovic, D., Schuster, B. S. & Shi, Z. Surface tension and viscosity of protein condensates quantified by micropipette aspiration. *Biophys. Reports* 1, (2021).

According to the Stokes-Einstein equation ($D = \frac{kT}{r \cdot 6 \eta}$), the diffusion coefficient D of a particle depends on the viscosity of a solution η , via $D \propto \eta^{-1}$. Thus, every order of magnitude that the viscosity is higher, the diffusion is lower. From the diffusion coefficient, we can determine the critical radius at which condensates can form double-emulsion condensates R_c , as explained in the main text via $D_{eff} = R_c^2$. Thus, while in the condensates in this paper we measure a diffusion coefficient of $12 \mu\text{m}^2/\text{s}$, the diffusion in biologically relevant systems can be orders of magnitude slower. The graph below, is a more extensive version to the one previously shown to the reviewer and has been included in the manuscript. We have determined the critical radius above which double-emulsion condensates can be formed at $D = 12, 1.2, 0.12$ and $0.012 \mu\text{m}^2/\text{s}$.

When the diffusion coefficient is 2 – 3 orders of magnitude lower than it is for the PolyA PEG condensates we have worked with, the critical size is 1 μm or less. Thus, given the orders of magnitude higher viscosity and thus orders of magnitude slower diffusion coefficient, our mechanism can help explain how this structure is formed in smaller and denser condensates found in cells as well. We appreciate that the reviewer has pointed out the importance of discussing this in the manuscript (changes in Ext. Data figure 10 and lines 176-180 and 383-385).

The reviewer also asks if we can make condensates of either just PolyA or just PEG as comparison. Sadly, PolyA and PEG do not form condensates in each other's absence. We understand that this is important to mention to the reader early in the manuscript to understand our model system. Thus, we have added the information that the compounds co-condensate in line 44. Reference 26, added last revision, also contains information about this condensate system.

2. In response to my comment 2, the authors have provided a plot showing the percentage of molecules at the surface given a condensate and particle radius. While this plot makes it clear that their condensates are large enough to ignore surface effects, when considering biologically relevant scales - such as the 10nm particle size and 0.1 micron condensate size example provided by the authors - the percent of molecules on the surface is between 10-20%. This is a significant portion and further calls into question the in vivo relevance of this work, given that at biologically relevant length scales the surface effects would be non-negligible. This is a critical point that needs to be addressed in the manuscript but is not. Again, neither the graph that authors show, nor any discussion of this point is mentioned anywhere in the manuscript.

Indeed, we expect 0.1 μm condensates containing particles of 100 nm in size to have 10-20% of the molecules at the surface. Condensates of approximately 300 nm can have this structure. Thus, we have considered two ways in which surface effects could potentially play a role in effecting the mechanism for double-emulsion formation in submicron condensates. Firstly, surfaces exert Laplace pressure which shift the dense phase concentrations slightly away from the equilibrium value. Thus, condensates are expected to be slightly denser very close to the surface. Notably, our mechanism depends on changes in the density. Not specific, but relative values are of importance. Since the slightly denser phase is still expected to change density upon environmental changes, we do not expect a significant change in the mechanism from this effect. The second effect is that molecules at the surface could have a slightly different conformation from those in the bulk of the liquid. Via a combination of experiments, referenced literature and computational model, we found that kinetically arrested phase separation is a general phenomenon, independent of the material of which condensates are made up. Thus, we don't expect slight conformational changes at different locations in the condensate to have a large affect on the mechanism. Notably, there is currently very little research available about how surface effects influence the dynamics of biomolecular condensates. We feel this is outside of the scope of this paper but find this a very interesting topic for future study.

3. It is unclear what the authors meant in their response to my comment 3. To better recreate the environment of the cell, they describe creating a liquid with slightly different compounds present in the surrounding dilute phase. It is unclear if these are future experiments or ones that have already been done. If it is the latter, then those results should be described further. Clarification is needed. In any case, the authors have addressed my point here through their response to another reviewer, where they changed their terminology from "multiphase" to "double emulsion". This suitably distinguishes their condensates from multiphase biological structures such as nucleoli.

The comment by the reviewer was misunderstood, apologies. Indeed, the term "multiphase", in which multiple dense phases result in a condensate with internal structure, has been removed and "double-emulsion" was used instead throughout the manuscript.

4. The authors have added a new figure illustrating and quantifying the effect of polyA addition to stress granules. While this is an interesting experiment using biological analogs, there is still a minimum radius needed to create the double emulsions, around 5 microns. This is still far larger than is biologically relevant, and no discussion is presented in the manuscript that addresses this point for potential readers.

In Figure 4 and Ext. Data Fig. 7, reconstituted stress granules can form this double-emulsion structure in vitro starting from approximately 5 μm , as the reviewer points out. In comment 1, we reference and discuss literature that shows that the viscosity of condensates prepared in vitro can be orders of magnitude smaller than that of condensates in cells. Given that the critical radius decreases significantly with a decrease in diffusion coefficient, it is thus likely that double-emulsion structures may arise in much smaller stress granules in vivo. We have added this information in line 179-180 (it is difficult to discuss this earlier in the manuscript, because the reader hasn't read about the importance of the diffusion rate / viscosity yet in the section about stress granules).

5. In my comment 5, I raise question about activity, which also was not addressed anywhere in the manuscript.

While activity does not influence the spinodal and binodal concentrations, it could cause kinetically arrested phase separation. A specific example is the formation of double-emulsion condensates when additional RNA is added to reconstituted stress granules. However, this was not added to the main text. Now, this can be found in lines 126-127.

6. In my comment 6, the authors made appropriate changes in the labelling of PolyA and PEG, but did not address if there is a critical size where polyA does not form cavities but PEG does.

Since PolyA and PEG do not form separate condensates, but only together, they form trapped droplets under the same conditions. The critical size is thus also the same. We hope that adding the information that they co-condensate early in the manuscript will prevent this misunderstanding for the readers and thank the reviewer for bringing this up again.

7. For comment 7, the authors stated that the PolyA used was polydisperse (700-3500kDa), making analysis of diffusion coefficients more difficult. While this does provide an explanation as to why the more biologically relevant molecule was not used, it does not provide any explanation as to why other slow diffusing compounds were not attempted. Given that using a compound with a diffusion rate closer to $1 \mu\text{m}^2/\text{s}$ would dramatically increase the biological relevance of this paper, an explanation should be included into the manuscript.

The authors also included simulations of rapid or slow composition changes. I think that these simulations could be used to extend this work's biological relevance, by simulating smaller condensates made up of compounds with slower diffusion rates. If double emulsion formation cannot be achieved under those simulation conditions, that should be addressed as it impacts the application of this model to smaller biological systems.

This model system was chosen as the PolyA PEG condensates do not significantly solidify over time, in contrast with many denser condensates made from proteins, like FUS, which do [1]. If the condensate did solidify, experiments like the one shown in figure 1A, where a condensate is observed over an extended period of time at high temperatures, would not be possible, since multiple variables would be changing at the same time.

Using condensates with a higher diffusion rate allowed us to observe trapped droplets in large condensates and more easily find the critical size. Additionally, we were able to observe fusion events on long enough time-scales to study them quantitatively (Figure 3). We have included references of this structure observed in cells and a model that shows that the critical size is only dependent on rate of composition change and diffusion rate (Ext. text 2, line 103-105)

[1] Patel, A. et al. A liquid-to-solid phase transition of the ALS protein FUS accelerated by disease mutation. Cell 2015.

We thank the reviewer again for their help with improving the manuscript.

Reviewer #4:

The authors have responded to all the comments and concerns I raised. Their responses and revisions exceed all reasonable expectations. This is important and timely work and should be published without further revisions. Nature Communications is the right forum for this work.

We thank the reviewer for their assessment of the manuscript and their fantastic feedback.

REVIEWER COMMENTS

Reviewer #3 (Remarks to the Author):

The authors have addressed many of the raised concerns, however, some points critical to the paper's impact remain unaddressed, which I list below. Especially, the question of biological relevance still remains open and not satisfactorily addressed in the paper's simulations and conclusions. The claims of biological relevance should be either supported by evidence or removed from the text. As of now, I cannot recommend publication in Nature Communications.

The remaining major concerns are:

1. The biologically relevant condensate sizes have still not been shown; authors provide only a speculation, but no evidence, that their model can account for biologically relevant sizes. The conclusions of the paper should reflect that. Moreover, there was no response from authors (in the response document or in the manuscript) to suggested simulations that could address this issue. Viscosity and diffusion rates are closely related, but whether the authors' model can successfully simulate such condensates is still unknown. Can the model explicitly predict smaller sized condensates when tuning the viscosity towards biological relevance? If so, including a graphic of those sizes would strengthen the paper considerably.
2. I agree with authors that investigating surface effects in detail is beyond the scope of this paper; nevertheless, surface effects should be listed and discussed as a potential caveat to the authors' model, especially when approaching biologically relevant size range. Presently, there is no mention of this in the manuscript.

Reviewer #3:

The authors have addressed many of the raised concerns, however, some points critical to the paper's impact remain unaddressed, which I list below. Especially, the question of biological relevance still remains open and not satisfactorily addressed in the paper's simulations and conclusions. The claims of biological relevance should be either supported by evidence or removed from the text. As of now, I cannot recommend publication in Nature Communications.

The remaining major concerns are:

1. The biologically relevant condensate sizes have still not been shown; authors provide only a speculation, but no evidence, that their model can account for biologically relevant sizes. The conclusions of the paper should reflect that. Moreover, there was no response from authors (in the response document or in the manuscript) to suggested simulations that could address this issue. Viscosity and diffusion rates are closely related, but whether the authors' model can successfully simulate such condensates is still unknown. Can the model explicitly predict smaller sized condensates when tuning the viscosity towards biological relevance? If so, including a graphic of those sizes would strengthen the paper considerably.

Previously, we have modelled the critical size for condensates to from double-emulsion condensates, depending on the diffusion rate and cooling rate (Ext. Data Fig. 10c). The reviewer points out that it would be great for the model to be adjusted to apply to biologically relevant condensates, specifically pointing out the importance of size. We have expanded our model to do just and provide a graph showing the results (Ext. Data Fig. 10d).

We determine the critical size as a function of rate of interaction energy change. This generalizes the model to look at required interaction energy changes, rather than cooling rate. Additionally, we can now find the exact requirements for condensates to obtain a double-emulsion structure, even for smaller condensates. The critical radius R changes as a function of rate of cooling v with the cooling effectively changing the interaction parameter χ , defined as $\chi \equiv \frac{\epsilon}{k_B T}$ in the Flory-Huggins theory, with $k_B T$ unit thermal energy and ϵ the interaction energy. The same change in χ , which we denote by $\delta\chi$, can be induced also by changing ϵ instead of T , and the two can be related using $\delta\chi = \frac{\partial\chi}{\partial\epsilon} \delta\epsilon = \frac{\partial\chi}{\partial T} \delta T$. This gives

$$\frac{1}{k_B T} \delta\epsilon = - \frac{\epsilon}{k_B T^2} k \delta T$$
$$\frac{\delta\epsilon}{\epsilon} = - \frac{\delta T}{T}$$

and the fractional change in T is proportional to the fractional change in ϵ for the same $\delta\chi$. Using the above we can use the rate of change of temperature to calculate an equivalent rate of change of ϵ . Additionally, we estimate $\epsilon \approx k_B T$, since the energy scale driving phase separation is of the order $k_B T$ (reference 1). Using these, we convert the rate of change of temperature v to the rate of change of interaction energy v_ϵ via

$$v = \frac{\delta T}{\delta t}$$
$$= - \frac{T}{\epsilon} \frac{\delta\epsilon}{\delta t}$$

$$= - \frac{300K}{k_B T} v_\epsilon$$

Combining this with our previously established model, we construct Ext. Data Fig. 10d:

Here, the blue line shows the critical radius above which double-emulsion condensates are formed and below which they are not formed. This is shown as a function of rate of change of interaction energy and modelled for condensate smaller than 1 μm in radius. We observe now that smaller condensates can form a double-emulsion structure, but that the interaction energy between the molecules in the condensate needs to change at very significant rates. This requirement is lower for larger condensates. Our model now simulates the requirements for smaller condensates to form a double-emulsion structure. Since we found smaller condensates to be less able to form this structure, we have incorporated this limitation into the manuscript (lines 178-180 and 394-397, Extended Data Fig. 10d and Supplementary text 5).

[1] Qian, D. Michaels, T.C.T., Knowles, T.P.J., Analytical solution to the Flory-Huggins Model. *J. Phys. Chem. Lett.* **2022**, 13, 33, 7853–7860

2. I agree with authors that investigating surface effects in detail is beyond the scope of this paper; nevertheless, surface effects should be listed and discussed as a potential caveat to the authors' model, especially when approaching biologically relevant size range. Presently, there is no mention of this in the manuscript.

We agree that this caveat should be stated in the manuscript. This limitation is described in lines 180-182.